# Wrong Model, Right Uncertainty: Spatial Associations for Discrete Data with Misspecification

## Abstract

Scientists are often interested in estimating an association between a covariate and a binary- or count-valued response. For instance, public health officials are interested in how much disease presence (a binary response per individual) varies as temperature or pollution (covariates) increases. Many existing methods can be used to estimate associations, and corresponding uncertainty intervals, but make unrealistic assumptions in the spatial domain. For instance, they incorrectly assume models are well-specified. Or they assume the training and target locations are i.i.d. — whereas in practice, these locations are often not even randomly sampled. Some recent work avoids these assumptions but works only for continuous responses with spatially constant noise. In the present work, we provide the first confidence intervals with guaranteed asymptotic nominal coverage for spatial associations given discrete responses, even under simultaneous model misspecification and nonrandom sampling of spatial locations. To do so, we demonstrate how to handle spatially varying noise, provide a novel proof of consistency for our proposed estimator, and use a delta method argument with a Lyapunov central limit theorem. We show empirically that standard approaches can produce unreliable confidence intervals and can even get the sign of an association wrong, while our method reliably provides correct coverage.

## 1 Introduction

Estimating associations between spatial variables and a binary- or count-valued response is fundamental across scientific disciplines. For instance, researchers are interested in (a) how much cardiovascular disease (a binary response per individual) increases with air pollution in Chinese cities (Zhao et al., 2015), (b) how the number of hospital admissions (a count-valued response per hospital) increases with temperature in European cities (Michelozzi et al., 2009), and (c) the extent to which ozone exceeding health guidance (a binary outcome) increases with meteorological variables in major cities in Texas (Vizuete et al., 2022). Moreover, quantifying uncertainty in these associations is fundamental for scientific and public health decision-making.

There are two natural approaches. (A) We might fit a highly flexible classifier — e.g., a transformer (Vaswani et al., 2017), or gradient-boosted tree (Chen and Guestrin, 2016) — and then apply a post hoc interpretability method (e.g. Lundberg and Lee, 2017; Ribeiro et al., 2016). But data in these applications are often very sparse in space, so we might hope to estimate an association well even when prediction quality could be very poor. (B) We might fit an interpretable model to start. For instance, when the response is continuous, Buja et al. (2019a) argue that a linear model can be used to estimate associations even when the data are highly nonlinear — that is, even when the linear model is (potentially very) misspecified.

Submitted to Reliable ML from Unreliable Data Workshop (NeurIPS 2025). Do not distribute.

Additional challenges arise in the applications described, though. Namely, the spatial locations where we want to draw inferences need not align well with the locations where we have data. E.g., in example (c) above, scientists have access to sensors across the state but are interested in associations in major Texas cities. Moreover, neither the training nor target locations are random. E.g., in Texas, air pollution monitor placement is decided by state and local governments under regulatory constraints from the United States Environmental Protection Agency. And major Texas cities are not randomly sampled from a larger population. For continuous responses, Burt et al. (2025a) address these concerns; they provide confidence intervals that maintain nominal coverage over spatial associations, even when training and target spatial locations can be nonaligned and nonrandom.

However, their method requires continuous responses with homoskedastic (spatially constant) noise. In all of the applications discussed above, and many other spatial analyses, the response is binary- or count-valued, and so the noise is heteroskedastic in space. To instead provide confidence intervals for binary- or count-valued data, we might naturally think to apply the delta method (van der Vaart, 1998, Chapter 3) to the estimator from Burt et al. (2025a). However, the delta method requires a consistent point estimate. In the present work, we show that the point estimate from Burt et al. (2025a) is *not* generally consistent.

Therefore, we need both a new estimator, as well as a new confidence interval, for the binary- and count-valued response setting. We provide these in the present work. Along the way, we also provide an estimator and asymptotically valid confidence intervals for continuous responses with spatially varying noise. In particular, we suggest a new point estimate inspired by Buja et al. (2019a), Buja et al. (2019b), Burt et al. (2025b), and Burt et al. (2025a); our estimate starts from a (misspecified but interpretable) generalized linear model (GLM) but takes into account nonrandom and nonaligned sampling of spatial locations. We show that in an *infill asymptotic setting*, where we have a sequence of spatial locations that eventually becomes dense in space but may not be sampled from any probability measure, our estimator is consistent. This consistency requires adaptivity; we demonstrate that the estimator of Burt et al. (2025a) and standard GLM point estimates using the training data are generally not consistent in this setting. We establish asymptotic normality of our estimator under conditions strictly more general than assuming training locations are sampled from a distribution supported around target locations. Our approach requires a Lyapunov central limit theorem applicable to non-identically distributed data. We propose a new, computationally efficient variance estimator suitable for problems with spatially varying noise and prove its consistency under infill asymptotics. Combining these results, we propose confidence intervals that can be computed efficiently from the available data, and prove that these confidence intervals are asymptotically conservative.

Our simulations demonstrate that existing methods can lead to fundamentally incorrect conclusions. In some cases, all baseline confidence intervals achieve zero empirical coverage and produce associations with the wrong sign while excluding zero. Our method consistently achieves coverage at or above the nominal level and never produces wrong-signed associations with confidence intervals excluding zero. Importantly, one simulation requires extrapolation, demonstrating that even when infill assumptions are unrealistic, our approach often provides conservative uncertainty estimates.

## 2   Setup and Background

We first describe our data and data-generating process. Then we describe our (misspecified) model and estimand. Our assumed data-generating process and estimand in this section are similar to those in Burt et al. (2025a). Our estimator, theory, and experiments form our major contributions (in subsequent sections) and are substantially different from Burt et al. (2025a).

### 2.1   Data-Generating Process

The training data consist of $N$ fully observed triples $(S_n, X_n, Y_n)_{n=1}^N$, with spatial location $S_n \in \mathcal{S}$, covariate $X_n \in \mathbb{R}^P$, and response $Y_n \in \mathcal{Y} \subset \mathbb{R}$. While our motivation and experiments focus on $\mathcal{Y} = \{0, 1\}$ (binary-valued) or $\mathcal{Y} = \mathbb{N}$ (count-valued), our treatment also handles $Y_n \in \mathbb{R}$. $\mathcal{S}$ represents geographic space; we assume $\mathcal{S}$ is a metric space with metric $d_{\mathcal{S}}$. We collect the training covariates in the matrix $X \in \mathbb{R}^{N \times P}$ and the training responses in the $N$-tuple $Y \in \mathcal{Y}^N$.

The target data consist of $M$ pairs $(S_m^\star, X_m^\star)_{m=1}^M$, with $S_m^\star \in \mathcal{S}$, $X_m^\star \in \mathbb{R}^P$. The corresponding responses $\{Y_m^\star\}_{m=1}^M$ are unobserved. We collect target covariates in $X^\star \in \mathbb{R}^{M \times P}$ and unobserved

target responses in a tuple $Y^\star \in \mathcal{Y}^M$. Our goal is to use the training data to estimate associations between covariates and responses at these new target locations.

**Similar assumptions to past work.** Our first three assumptions follow Burt et al. (2025a) in allowing a smooth, nonparametric relationship between spatially varying variables. We start by assuming that both training and target covariates are fixed functions of spatial location. This assumption is most natural when covariates represent environmental or meteorological measurements taken at specific times, or averaged over a time period.

**Assumption 1** (Burt et al. (2025a), Assumption 1)**.** *There exists a (deterministic) function* $\chi : \mathcal{S} \to \mathbb{R}^P$ *such that* $X_m^\star = \chi(S_m^\star)$ *for* $1 \le m \le M$ *and* $X_n = \chi(S_n)$ *for* $1 \le n \le N$.

As in Burt et al. (2025a), we assume that the conditional expectation of the response can be written as $\mathbb{E}[Y_n|X_n, S_n] = g(X_n, S_n)$, for some nonparametric function $g$. Under Assumption 1, the covariates are themselves fixed functions of location, so we can define $f : \mathcal{S} \to \mathbb{R}$, $f(S) = g(\chi(S), S)$. In other words, $f$ maps each spatial location directly to the expected value of the response at that location. Importantly, unlike Burt et al. (2025a, Assumption 2), we do not assume homoskedastic, Gaussian noise; we instead allow spatially varying noise and discrete response variables.

**Assumption 2.** *There exists a function* $f : \mathcal{S} \to \mathbb{R}$ *such that for all* $m \in \{1, \ldots, M\}, \mathbb{E}[Y_m^\star|S_m^\star] = f(S_m^\star)$ *and for all* $n \in \{1, \ldots, N\}, \mathbb{E}[Y_n|S_n] = f(S_n)$. *Moreover,* $Y_m^\star|S_m^\star$ *and* $Y_n|S_n$ *are independent for all* $1 \le m \le M$ *and* $1 \le n \le N$.

Assumption 3 encodes the idea that nearby points in space have similar expected responses. Intuitively, it rules out arbitrarily sharp changes in $f$ across very small spatial distances. This pattern is common in environmental and geostatistical data, where smooth spatial variation is a natural prior belief.

**Assumption 3** (Burt et al. 2025a, Assumption 4)**.** *The conditional expectation of the response,* $f$, *is an* $L$-*Lipschitz function from* $(\mathcal{S}, d_\mathcal{S}) \to (\mathbb{R}, |\cdot|)$. *That is, for any* $s, s' \in \mathcal{S}, |f(s) - f(s')| \le Ld_\mathcal{S}(s, s')$.

**New data-generating process assumptions.** Because we do not assume spatially constant Gaussian errors on the responses, we need assumptions that control the tail behavior of the possible responses. Our next three assumptions concern higher moments of the response as a function of spatial location. Specifically, we assume that we can define a conditional variance function and a conditional fourth central moment function, and that these functions are bounded (and, for the variance, continuous). These conditions are generally quite mild. For binary responses, these assumptions hold automatically: the variance is bounded because the outcome is bounded, and continuity of the mean (from Assumption 3) already implies continuity of the variance. For count and continuous responses, it is natural to expect that the probability mass or density of the outcome varies smoothly across space. This intuition is even stronger than required here, since smoothness of the probability distribution implies continuity of the variance. Finally, for any uniformly bounded response, both the bounded variance (Assumption 4) and bounded fourth moment (Assumption 6) conditions follow immediately.

**Assumption 4.** *There exists a conditional variance function* $\rho^2 : \mathcal{S} \to [0, \infty)$ *defined by* $\rho^2(s) = \mathbb{E}[(Y(S) - f(S))^2|S = s]$, *and this function is uniformly bounded by a constant* $B_Y$.

**Assumption 5.** *The function* $\rho^2$ *from Assumption 4 is continuous on* $\mathcal{S}$.

**Assumption 6.** *There exists a conditional fourth central moment function* $\alpha : \mathcal{S} \to [0, \infty)$ *defined by* $\alpha(s) = \mathbb{E}[(Y(S) - f(S))^4|S = s]$, *and this function is uniformly bounded by a constant* $C$.

## 2.2 Model and Estimand

Generalized linear model coefficients describe the direction and magnitude of the associations between covariates and discrete response variables, and will be our inferential target. A (well-specified) GLM assumes that — for a covariate-response pair $(x, y)$ — the distribution of the response $y$ has probability mass function $h(y; \theta) = c(y) \exp(\theta y - \kappa(\theta)), \theta = x^\mathrm{T}\beta^\star$ (Nelder and Wedderburn, 1972; McCullagh and Nelder, 1989) where $\theta$ is the canonical parameter, $\kappa$ is the cumulant generating function, $c(y)$ is a base measure, and $\beta^\star$ are the true regression coefficients. $\kappa$ is convex and infinitely differentiable. The data log-likelihood is

$$\ell(\beta; Y) = C + \sum_{n=1}^{N} X_n^\mathrm{T}\beta Y_n - \kappa(X_n^\mathrm{T}\beta), \tag{1}$$

where $C$ is a term that does not depend on $\beta$. Under a well-specified model with independent and identically distributed (i.i.d.) data and mild regularity conditions, the maximum likelihood estimator obtained by maximizing Eq. (1) converges to the true coefficients $\beta^\star$ (Wald, 1949). In contrast, when the model is misspecified, maximizing the log-likelihood instead yields the coefficients that minimize the Kullback–Leibler (KL) divergence between the model and the true data-generating process (White, 1982). In either case, the estimator is asymptotically normal. We discuss the use of asymptotic normality to construct confidence intervals for parameters in well-specified GLMs, as well as other approaches for constructing confidence intervals in GLMs in Appendix B.

**Our Maximum Likelihood Estimand.** Our goal is to describe how covariates are associated with the response variable at the target locations, using data observed at the training locations. Because these two sets of locations may differ, we define our estimand as the parameter in the (parametric) GLM family considered that provides the best approximation to the true response process at the target distribution of locations. This generalizes the least squares approach considered in Burt et al. (2025a) to other (non-Gaussian) exponential families and follows the general framework of fitting parametric models as 'projections' outlined in Buja et al. (2019b, §2.1). Formally, we define the population maximum likelihood parameter conditional on the target locations as

$$\beta^{\mathrm{MLE}} = \arg\max_{\beta \in \mathbb{R}^P} \sum_{m=1}^{M} \mathbb{E}[\log h(Y_m^\star; X_m^{\star\mathrm{T}}\beta)|S_m^\star]. \tag{2}$$

In Appendix A, we show that $\beta^{\mathrm{MLE}}$ equivalently minimizes the Kullback–Leibler divergence between the data-generating process and the GLM family, conditional on the distribution over locations taken to be the target distribution.

**Assumption 7.** *There exists a parameter $\beta^{MLE}$ solving Eq. (2), and the corresponding population log-likelihood is strictly concave in an open neighborhood containing $\beta^{MLE}$.*

Assumption 7 guarantees uniqueness of the estimator and ensures that the Hessian of the log-likelihood is positive definite at $\beta^{\mathrm{MLE}}$. In the case of linear models, a necessary and sufficient condition is that $X^\star$ is full-rank (c.f. Burt et al., 2025a, Assumption 4). More generally, it is necessary that $X^\star$ is full rank, though not always sufficient. Intuitively, this condition prevents attempting to estimate more parameters than there are independent pieces of information at the target sites. In what follows, we focus on inference — both point estimates and confidence intervals — for individual parameters of interest, $\beta_p^{\mathrm{MLE}} = e_p^{\mathrm{T}}\beta^{\mathrm{MLE}}$, where $e_p \in \mathbb{R}^P$ is the unit vector selecting the $p$th component (i.e., with a single 1 at entry $p$ and 0 elsewhere).

# 3 Inference for Misspecified GLMs Under Infill Asymptotics

In this section, we describe our procedure for inference in generalized linear models with misspecification and nonrandom spatial sampling.

**Overview of Inference Strategy.** A desirable property for an estimator is consistency: with enough training data, the estimator should converge to the estimand, the true underlying quantity of interest. In our spatial setting, however, it is not just the amount of training data that matters, but also where the data are located. This naturally leads to the framework of *infill asymptotics*, which considers the case where increasingly many training points are observed in the neighborhoods of the fixed target locations. In Section 3.1, we show that existing methods are not necessarily consistent even in this idealized setting, and propose an estimator that is. While estimating an association consistently is reassuring for many scientific applications, it is also important to quantify uncertainty about the quality of this point estimate. In Section 3.2, we use a Lyapunov central limit theorem (for non-identically distributed data) to show our point estimate is asymptotically normal. This allows us to construct confidence intervals around our point estimate that are asymptotically valid. These confidence intervals depend on the (unknown) variance of the response at the target locations. We propose a computationally efficient estimator for this spatially varying variance, and prove its consistency under infill asymptotics.

## 3.1 Consistency under Infill Asymptotics

We adopt the infill asymptotic framework of, e.g., Cressie (2015, §5.8) and Burt et al. (2025b, §3).

**Definition 1** (Infill Asymptotics). *Given a (fixed) set of target locations $(S_m^\star)_{m=1}^M$, a sequence of training locations $(S_n)_{n=1}^\infty$ satisfies infill asymptotics with respect to $(S_m^\star)_{m=1}^M$ if, for all $1 \leq m \leq M$, and any open neighborhood $U_m$ containing $S_m^\star$, $|\{n \in \mathbb{N} : S_n \in U_m\}| = \infty$.*

Intuitively, infill asymptotics requires that around each target location, the training set becomes arbitrarily dense as the sample size grows. In Appendix C we give an example showing that even under favorable conditions — Gaussian noise and smooth response surface — both the point estimate based on 1-nearest-neighbor considered in Burt et al. (2025a) and the ordinary least squares estimate can fail to achieve consistency under infill asymptotics with model misspecification.

**A Consistent Estimator under Infill Asymptotics.** We develop an estimator that is consistent under infill asymptotics. Our approach builds on the intuition of Burt et al. (2025a), who proposed borrowing training responses to estimate (unobserved) responses at target locations. However, the key modification we introduce to ensure consistency is to allow the number of neighbors used for borrowing to grow adaptively with the size of the training set. Burt et al. (2025b) relied on a similar adaptive construction to show consistency in the simpler setting of mean estimation.

Define the function $\tau : \mathbb{R}^M \to \mathbb{R}^P$, $\tau(A) = \arg\max_{\beta \in \mathbb{R}^P} \sum_{m=1}^M X_m^{\star\mathrm{T}} \beta A_m - \kappa(X_m^{\star\mathrm{T}} \beta)$. The estimand (Eqs. (1) and (2)) is $\beta^{\mathrm{MLE}} = \tau(\mathbb{E}[Y^\star|S^\star])$. Our strategy is to average information from responses near each target point to build an estimator, $\hat{A}$, for $\mathbb{E}[Y^\star|S^\star]$. And then to use $\tau(\hat{A})$ as an estimator for $\beta^{\mathrm{MLE}}$. To instantiate this, we follow Burt et al. (2025a, Definition 10) and use a nearest-neighbor weighting scheme.

**Definition 2** (Nearest-Neighbor Weight Matrix). *Given training locations $(S_n)_{n=1}^N$, target locations $(S_m^\star)_{m=1}^M$, and a fixed $k_N \in \mathbb{N}$, define the $k_N$-nearest-neighbor weight matrix by*

$$\Psi_{mn}^{N,k_N} = \begin{cases} 1/k_N & S_n \in \{k_N \text{ closest training locations to } S_m^\star\} \\ 0 & \text{otherwise.} \end{cases} \tag{3}$$

*For definiteness, we assume that, if multiple training locations are equidistant from a target, ties are broken uniformly at random.*

This yields an estimator that we can calculate from the observed data:

$$\widehat{\beta}^{N,k_N} = \tau(\Psi^{N,k_N} Y) = \arg\max_{\beta \in \mathbb{R}^P} \sum_{m=1}^M X_m^{\star\mathrm{T}} \beta (\Psi^{N,k_N} Y)_m - \kappa(X_m^{\star\mathrm{T}} \beta). \tag{4}$$

Burt et al. (2025a) proposed the same estimator with $k_N = 1$, so that each target location borrows information only from its closest training neighbor. While this approach may be adequate empirically when the number of target locations is large, Counterexample 1 shows that it fails to deliver consistency under infill asymptotics. Since consistency of $\hat{\beta}_N$ is a prerequisite for establishing asymptotic normality of our estimator, a more robust choice of $k_N$ is required. We propose an adaptive rule for selecting $k_N$: the key idea is to gradually increase the number of neighbors whenever the current neighbors (including the newly observed training location) are all sufficiently close to the target sites.

**Theorem 1.** *Fix any $M \in \mathbb{N}$ and $(S_m^\star)_{m=1}^M$. Let $(S_n)_{n=1}^\infty$ be a sequence of points in $\mathcal{S}$ such that infill asymptotics holds with respect to $(S_m^\star)_{m=1}^M$. Suppose Assumptions 1 to 4 and 7. With adaptively chosen neighbors as discussed in Theorem D.1, $\widehat{\beta}^{N,k_N} \to \beta^{MLE}$, where convergence is in probability.*

The proof of Theorem 1 as well as a formal characterization of the adaptive scheme for selecting the number of neighbors are provided in Appendix D. Intuitively, the procedure adapts the number of neighbors so that as training data accumulate near the targets, the estimator gradually incorporates more information without sacrificing local accuracy.

**Limitations when Extrapolating.** In cases where extrapolation is needed because the training data are not available near the target locations (either because of finite data or because the distribution of training locations is not supported near the target locations), we cannot hope to estimate $\beta^{\mathrm{MLE}}$ arbitrarily well. In particular, we simply do not know how $\mathbb{E}[Y_m^\star|S_m^\star]$ behaves in the extrapolation setting, and our assumptions together with the data are not strong enough for $\beta^{\mathrm{MLE}}$ to be identified. Our approach therefore focuses on the regime where infill asymptotics holds, which is precisely the setting where consistent estimation is achievable.

## 3.2 Asymptotically Valid Confidence Intervals

We now focus on quantifying uncertainty around $\widehat{\beta}^{N,k_N}$. Our focus is on the construction of confidence intervals that are (asymptotically) guaranteed to achieve nominal coverage. Precisely, we will construct confidence intervals that satisfy the following under our data generating assumptions and infill asymptotics.

**Definition 3** (Asymptotically Conservative Confidence Interval). *For any $1 \leq p \leq P$ and any $\alpha \in (0,1)$ a sequence of confidence intervals $(I_{p,N}^\alpha)_{N=1}^\infty$ is asymptotically conservative if $\lim_{N \to \infty} \mathbb{P}(\beta_p^{MLE} \in I_{p,N}^\alpha) \geq 1 - \alpha$.*

**Asymptotic Normality.** Constructing confidence intervals for arbitrary random variables is challenging. But constructing confidence intervals for normal random variables is easier, and so we follow a classical approach to deriving confidence intervals in which we first show that our estimator is asymptotically normal. We use a Lyapunov central limit theorem together with the delta method (van der Vaart, 1998, Chapter 3), to show that under the same setup as Theorem 1

$$\sqrt{k_N}(\beta^{\mathrm{MLE}} - \widehat{\beta}^{N,k_N}) \to \mathcal{N}(B, \tau'(\mathbb{E}[Y^\star|S^\star])^{\mathrm{T}} \Lambda^\star \tau'(\mathbb{E}[Y^\star|S^\star])), \tag{5}$$

$$B = \tau'(\mathbb{E}[Y^\star|S^\star])^{\mathrm{T}}(\mathbb{E}[Y^\star|S^\star] - \Psi^{N,k_N}\mathbb{E}[Y|S]) \quad \text{and} \quad \Lambda_{mm'}^\star = \delta_{mm'}\mathbb{V}[Y_m^\star|S_m^\star].$$

Here $\tau'$ maps from a point in $\mathbb{R}^M$ to the Jacobian of $\tau$ at that point. and $\delta_{mm'}$ is a Kronecker delta, so $\Lambda^\star$ is diagonal. A formal statement and proof are in Appendix E.4, Theorem E.2. Equation (5) depends on $\tau'(\mathbb{E}[Y^\star|S^\star])$, which is not observed. In practice and in our later theory, we use a (consistent) point estimate for this Jacobian $\tau'(\Psi^{N,k_N}Y)$.

**Bounding the Bias.** We need to control the bias, $B$. After replacing $\mathbb{E}[Y^\star|S^\star]$ with $\Psi^{N,k_N}Y$ each coordinate of the bias is a linear combination of evaluations of the conditional expectation of the response, $f$, at training and target locations. Burt et al. (2025a, Appendix B.2) showed that such a linear combination can be bounded in terms of a 1-Wasserstein distance that is efficiently computable. We provide additional detail in Proposition E.2.

**Plug-in Estimate of $\mathbb{V}[Y^\star|S^\star]$.** We do not have access to $\mathbb{V}[Y_m^\star|S_m^\star]$ for $1 \leq m \leq M$, which is needed to compute the variance of the point estimate. We propose a nearest-neighbor approach.

**Definition 4** (Nearest-Neighbor Variance Estimator). *For each $1 \leq n \leq N$, let $\zeta^N(n')$ be the index of the nearest-neighbor of $S_{n'}$ in the other training data $(S_n)_{n=1,n\neq n'}^N$. Define the diagonal matrix $\Lambda^N \in \mathbb{R}^{N \times N}$, $\Lambda_{nn}^N = \frac{1}{2}(Y_n - Y_{\zeta^N(n)})^2$.*

We show in Appendix E.2 that, assuming infill asymptotics, $k_N \Psi^{N,k_N} \Lambda^N \Psi^{N,k_N\mathrm{T}} \to \Lambda^\star$. Burt et al. (2025a) proposed to use $\frac{1}{N}\mathrm{tr}(\Lambda^N)$ to estimate the noise variance in homoskedastic linear regression, but did not establish its consistency or propose how to handle spatially varying noise.

**Statement of Confidence Intervals.** We now have the ingredients to define our confidence interval:

$$I_{p,N}^\alpha = \left[\widehat{\beta}_p^{N,k_N} - z_{\alpha/2}\hat{\sigma}_p - \tilde{B}_p, \widehat{\beta}_p^{N,k_N} + z_{\alpha/2}\hat{\sigma}_p + \tilde{B}_p\right], \tag{6}$$

with $\hat{\sigma}_p = \|(\Lambda^N)^{1/2}(\Psi^{N,k_N})^{\mathrm{T}}\tau'(\Psi^{N,k_N}Y)e_p\|_2$, $\tilde{B}_p = L \sup_{f \in \mathcal{F}_1} \left|\sum_{n=1}^N v_n^N f(S_n) - \sum_{m=1}^M w_m^N f(S_m^\star)\right|$,

Here $z_{\alpha/2}$ is the $(1 - \alpha/2)$ quantile of the standard normal distribution; $e_p \in \mathbb{R}^P$ is the $p$th standard basis vector; $w^N = X^\star \tau'(\Psi^{N,k_N}Y)e_p$; and $v^N = \Psi^{N,k_N}w^N$. The set $\mathcal{F}_1$ denotes the 1-Lipschitz functions on $(\mathcal{S}, d_\mathcal{S})$. We use $\|\cdot\|_2$ for the Euclidean ($\ell_2$) norm on vectors. A classic confidence interval $[\widehat{\beta}_p^{N,k_N} \pm z_{\alpha/2}\tilde{\sigma}_p]$ uses model-trusting standard errors and does not account for potential bias due to model-misspecification and nonrandom sampling. Sandwich estimators use standard errors that are valid under misspecification, but still do not account for potential bias because of the interaction between misspecification and nonrandom sampling. Our confidence interval, Eq. (6) uses standard errors that are still valid under misspecification, and accounts for potential bias.

**Asymptotic Validity of Confidence Intervals.** We now state our main result, that the confidence interval in Eq. (6) is conservative under infill asymptotics. We prove Theorem 2 in Appendix E.

**Theorem 2.** *Take the setup and assumptions of Theorem 1. Suppose the number of neighbors is chosen as in Theorem D.1 with $a_t = \frac{1}{\sqrt{t}}$ for $t \in \mathbb{N}$ and Assumptions 5 and 6. Then the confidence interval defined in Eq. (6) is asymptotically conservative.*

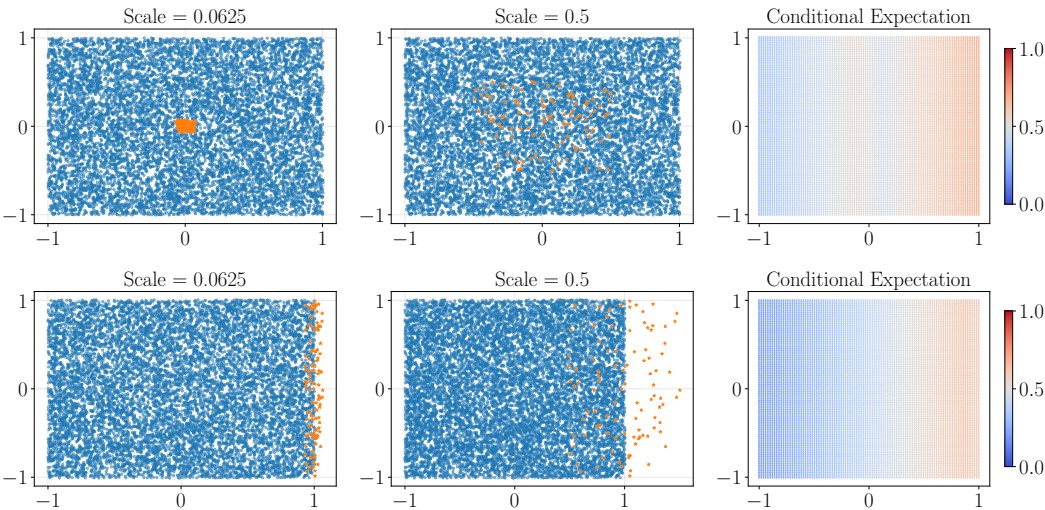

Figure 1: We summarize the data generating processes for the first (top) and second (bottom) simulation study. The left two plots show the distribution of train (blue) and target (orange) locations. The third panel shows the (unobserved) expected response surface.

## 4  Experiments

In this section, we present two simulation studies to evaluate the performance of the proposed method for logistic regression. Throughout, we consider three baselines: logistic regression, logistic regression using the sandwich covariance estimator (Huber, 1967), and weighted logistic regression using kernel density estimation (Shimodaira, 2000). While logistic regression is a classic method, confidence intervals from logistic regression are widely used in scientific applications (e.g. Lee et al., 2025; Zhang et al., 2023; Ahn et al., 2024). We give more detail on baseline methods in Appendix F.1.

**Evaluation Metrics.** We evaluate methods along four complementary dimensions. Our primary focus is on empirical coverage and the proportion of false positives, since failure on either dimension undermines the reliability of statistical conclusions. Empirical coverage measures the proportion of confidence intervals that contain the true parameter value; we regard a method as successful if its coverage is at or above the nominal level of $0.95$. The proportion of false positives measures the frequency with which a confidence interval excludes $0$ but assigns the wrong sign to the parameter; this rate should remain close to or below the nominal level of $0.05$. Conditional on reliability, we then assess whether methods provide informative conclusions. Two metrics capture this aspect: the average width of confidence intervals, which should be as small as possible given adequate coverage, and the proportion of true positives, defined as the fraction of intervals excluding $0$ with the correct sign, which should be as high as possible. Narrow intervals and a high rate of true positives indicate that a method can identify associations precisely and with confidence.

These metrics illustrate the balance between validity and informativeness. A method that always returns a degenerate interval of width zero (a single point) would appear confident whenever it guesses the correct sign, yet would completely fail to reflect uncertainty. Conversely, a method that always returns the entire real line would achieve perfect coverage and no false positives, but would provide no useful scientific guidance. We therefore regard a method as successful if it achieves coverage near the nominal rate, maintains a low false positive proportion, and produces intervals that are narrow enough to support meaningful conclusions — for example, correctly and confidently identifying the direction of association.

**Data-Generating Process.** In both simulations, we simulate 250 datasets according to data-generating processes described in detail in Appendix F.2 and illustrated in Fig. 1. The two simulations are intended to highlight contrasting regimes: the first one reflects a setting where the infill asymptotics assumption is reasonable, whereas for the second one extrapolation is unavoidable. In the latter case, we anticipate wider confidence intervals, reflecting the inherent difficulty of the task. For our method,

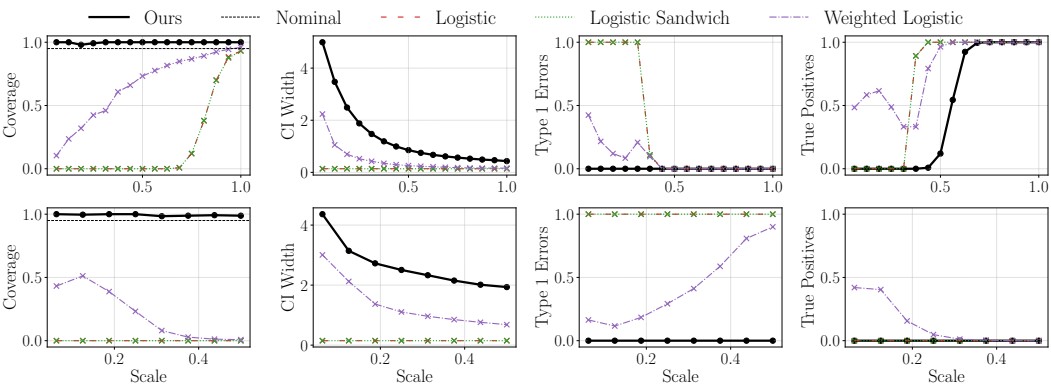

Figure 2: From left to right, coverage average confidence interval width, proportion of false positives and proportion of true positives for each method on the first simulation (top) and the second simulation (bottom). Coverage should be above the nominal level (dashed line in first column), and the proportion of false positives should be below $0.05$. Given these properties, we would like confidence intervals that are as narrow as possible, and return many true positives.

we set the Lipschitz constant of the conditional expectation function to its true value, $L = 0.25$, in both simulations.

In each experiment, we draw 10000 training locations uniformly from $[-1, 1]^2$. The target locations are then constructed differently across the two designs. In the first experiment, targets are concentrated within a subset of the square, determined by a scale parameter, so that the infill property holds. In the second experiment, targets are concentrated but shifted outside the main support of the training set, to the right of the square, thereby requiring extrapolation. The two left panels of Fig. 1 depict the distribution of training and target locations for the infill (top) and extrapolation (bottom) settings. In both experiments, we use a single covariate equal to the first coordinate of the spatial location. Responses are generated from a Bernoulli distribution whose conditional expectation varies smoothly with space. The rightmost panel of Fig. 1 displays this conditional expectation for both designs, with the precise mathematical forms given in Appendix F.

**Results.** We summarize the results across the two simulations in Fig. 2. Our method consistently achieves coverage at or above the nominal $0.95$ level and does not produce false positives. By contrast, the baseline methods frequently fall far short of nominal coverage: in the second simulation, all baselines achieve zero coverage for certain instances. This failure is accompanied by high rates of false positives, meaning the baselines often return intervals that confidently — but incorrectly — assign the wrong sign to the association.

The strength of our method lies in its reliability: it avoids misleading conclusions even in challenging extrapolation regimes. The cost of this conservativeness is wider confidence intervals and, consequently, a smaller proportion of true positives compared to the baselines. This trade-off is expected, as our method protects against worst-case bias rather than optimizing for power. Improvements in power may be possible, but in scenarios dominated by extrapolation, additional assumptions would be needed to confidently and correctly make inference about the direction of an association.

## 5   Discussion

In this work, we developed a new framework for inference on associations in generalized linear models under spatial misspecification and covariate shift. Through theory and simulations, we show that our estimator is consistent under infill asymptotics and that our intervals achieve valid coverage, unlike existing approaches which often fail dramatically. Our method is conservative, avoiding false positives even in challenging extrapolation settings. Looking ahead, we are particularly interested in applying our method to real datasets in scientific domains such as environmental monitoring, epidemiology, and climate science, where robust and reliable inference on spatial associations is critical.

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

404     air pollution and cardiovascular mortality in China: A systematic review and meta-analysis.
405     *Oncotarget*, 8.

## A   Interpretation of Target Maximum Likelihood

In this section, we show that Eq. (2) minimizes the conditional KL divergence from the true data-generating process over the model class, when the target locations are distributed according to the discrete measure that assigns equal weight to each target location. This follows the standard argument that maximum likelihood minimizes a KL divergence, but we reconstruct the argument to emphasize that in our setting it is conditional on the target locations.

**Proposition A.1.** *Suppose Assumptions 1, 2 and 7. Let $P^\star$ denote the joint measure of spatial locations, covariates and responses, with the measure over spatial locations fixed to equal the discrete measure that assigns equal weight to each target location. For $\beta \in \mathbb{R}^P$, define $P^\beta$ to be the measure over spatial locations fixed to equal the discrete measure that assigns equal weight to each target location, the covariates equal to $\chi(S)$, and the response generated with conditional log likelihood of the response equal to Eq. (1). Suppose there exists a $\beta \in \mathbb{R}^P$ such that $\mathrm{KL}(P^\star, P^\beta) \leq \infty$. Then, $\beta^{MLE} = \arg\min_{\beta \in \mathbb{R}^P} \mathrm{KL}(P^\star, P^\beta)$.*

*Proof.* Let $\Omega = \{\beta \in \mathbb{R}^P : \mathrm{KL}(P^\star, P^\beta) < \infty\}$. $\Omega$ is non-empty by assumption. And the minimizer of $\mathrm{KL}(P^\star, P^\beta)$ must occur in $\beta$ as this KL divergence is infinite outside of $\Omega$ by definition. Let $P^\star_{Y^\star_m | S^\star_m}$ denote the conditional distribution of $Y^\star_m$ given $S^\star_m$ under the data generating process, and $P^\beta_{Y^\star_m | S^\star_m}$ denote the conditional distribution of $Y^\star_m$ given $S^\star_m$ under the generalized linear model with parameter $\beta$. For any $\beta \in \Omega$, and using the chain rule of KL divergence (Cover and Thomas, 2006, Theorem 2.5.3), and because the measure of $P^\beta$ and $P^\star$ over the locations and covariates is the same by construction,

$$\mathrm{KL}(P^\star, P^\beta) = \frac{1}{M} \sum_{m=1}^{M} \int \log \frac{\mathrm{d}P^\star_{Y^\star_m | S^\star_m}}{\mathrm{d}P^\beta_{Y^\star_m | S^\star_m}} \mathrm{d}P^\star_{Y^\star_m | S^\star_m} \tag{A.1}$$

$$= \frac{1}{M} \sum_{m=1}^{M} \mathbb{E}[-\log h(Y^\star_m; X^{\star\mathrm{T}}_m \beta) | S^\star_m] + C, \tag{A.2}$$

where $C$ is the entropy (for discrete $Y$) or differential entropy (for continuous $Y$). Minimizing over $\beta$

$$\arg\min_{\beta\in\mathbb{R}^P}\mathrm{KL}(P^\star, P^\beta) = \arg\min_{\beta\in\Omega}\mathrm{KL}(P^\star, P^\beta) = \arg\max_{\beta\in\mathbb{R}^P}\sum_{m=1}^{M}\mathbb{E}[\log h(Y_m^\star; X_m^{\star\mathrm{T}}\beta)|S_m^\star], \quad \text{(A.3)}$$

The right hand side is the same as Eq. (2), and so $\beta^{\mathrm{MLE}}$ minimizes a KL divergence to the true data generating process, conditional on the target locations. □

# B  Alternative Approaches for Confidence Intervals for Well-Specified Generalized Linear Models

**Confidence Intervals Based on Asymptotic Normality.** A standard approach for constructing confidence intervals that are valid for large sample sizes follows from the general theory of asymptotic normality of maximum likelihood estimators (MLEs) Cramér (1946); Wald (1949). Informally, if $\hat{\beta}_n$ is the MLE of $\beta^\star$ based on $n$ samples, then under well-specification, $\sqrt{n}(\beta^\star - \hat{\beta}_n) \approx \mathcal{N}(0, I_{\beta^\star}^{-1})$ where $I_{\beta^\star}$ is the Fisher information matrix. In practice, $I_{\beta^\star}$ can be estimated using the observed Fisher information matrix, $(\hat{I}_{\beta,n})_{i,j} = \sum_{n=1}^{N}\frac{\partial^2\ell_n(\beta, Y_n)}{\partial\beta_i\partial\beta_j}$, where $\ell_n(\beta; Y_n) = C_n(Y_n) + X_n^{\mathrm{T}}\beta Y_n - \kappa(X_n^{\mathrm{T}}\beta)$ is the log-likelihood of a single data point. An asymptotic confidence interval for the $p$th coefficient $\beta_p^\star$ then takes the form: $\beta_p^\star \in \hat{\beta}_p \pm z_{1-\alpha/2}\tilde{\sigma}_p^2$, where $\tilde{\sigma}_p^2$ is the $p$th diagonal entry of $\hat{I}_{\beta,n}^{-1}$, and $z_{1-\alpha/2}$ is the $(1-\alpha/2)$-quantile of the standard normal distribution. Even if the model is misspecified, maximum likelihood leads to an asymptotically normal estimator when the data remain i.i.d., though the variance is no longer governed by the Fisher information. In this case, confidence intervals are obtained using a sandwich variance estimator (White, 1982). A detailed treatment of these asymptotics can be found in van der Vaart (1998, Chapter 4). We provide further discussion of alternative confidence interval constructions for well-specified GLMs in Appendix B.

**Alternative Approaches for Confidence Intervals in GLMs.** While the asymptotic approximation based on the observed Fisher information, described in Section 2, is widely used, there are other approaches exist for constructing confidence intervals for well-specified generalized linear models.

For logistic regression (Cox and Snell, 1989, Chapter 2) describes how to construct confidence intervals that are exact in finite samples. These exact methods are typically more computationally intensive, but can be used to construct confidence intervals that are valid even for small sample sizes.

Venzon and Moolgavkar (1988) use the asymptotic $\chi^2$ distribution of the profile log likelihood to construct asymptotic confidence intervals. The extent to which our methods can be adapted to these approaches is an interesting question for future work.

# C  Inconsistency of Point Estimation for Existing Methods

In this section, we provide additional details proving the claims in Counterexample 1. We first state the counterexample.

**Counterexample 1** (Several Existing Methods are Not Consistent Under Infill Asymptotics for Homoskedastic Linear Models with Gaussian Noise)**.** *Assume Assumptions 2 and 3 with spatial domain $[-0.75, 1]$, two target locations $S_m^\star = \pm 0.5$, $f(S) = S^2$ and $\chi(S) = S$. Suppose responses follow $Y^\star = f(S^\star) + \epsilon$, $\epsilon \sim \mathcal{N}(0, 1)$. Consider least squares linear regression fit without an intercept. Then Assumptions 4 to 6 hold, as does Assumption 7 with $\beta^{MLE} = 0$. Further, if the training data are uniformly distributed on $[-0.75, 1]$, then infill asymptotics holds almost surely. However, neither the estimator proposed in Burt et al. (2025a) nor the ordinary least square estimator based on the training data converge to $0$ in probability.*

The first claim we show is that Assumptions 4 to 6 and Assumption 7 hold, with $\beta^{\mathrm{MLE}} = 0$. First, $\rho^2(S) = \mathbb{V}(\epsilon) = 1$, and so Assumptions 4 and 5 hold. Next, the conditional 4th moment is again a constant function of space that is equal to the 4th moment of $\mathcal{N}(0, 1)$, which is 3, and is therefore bounded so Assumption 6 holds. Finally, the log likelihood is

$$\ell(\beta) = C + \frac{1}{2}\mathbb{E}[-(0.25 + \epsilon_1 + 0.5\beta)^2 - (0.25 + \epsilon_2 - 0.5\beta)^2] \quad \text{(C.1)}$$

Taking derivatives

$$\ell'(\beta) = \frac{1}{2}\mathbb{E}[-(0.25 + \epsilon_1 + 0.5\beta) + (0.25 + \epsilon_2 - 0.5\beta)] = -0.25\beta, \ell''(\beta) = -0.25. \quad \text{(C.2)}$$

This is (globally) concave by the 2nd derivative test, and has a unique maximum at the solution of $\ell'(\beta) = 0$, which is $\beta = 0$.

Our remaining claim is that OLS and the nearest-neighbor method with a single neighbor approach considered in Burt et al. (2025a) are not consistent. The ordinary least squares estimate converges to the solution of the training normal equations,

$$\mathbb{E}[x^2]^{-1}\mathbb{E}[xy] = \mathbb{E}[x^2]^{-1}\mathbb{E}[x^3] \neq 0, \quad \text{(C.3)}$$

where we used that because the distribution of $X$ is not symmetric about $0$, $\mathbb{E}[x^3] \neq 0$.

To show that estimator in Burt et al. (2025a) is not consistent, we show its variance does not converge to $0$. Because the distribution of $S^\star$ is absolutely continuous with respect to Lebesgue measure, with probability 1, for every $N$, there is a single training location closest to $S_1^\star$ and a single training location closest to $S_2^\star$. For all $N$, the variance of the estimator in Burt et al. (2025a) is then $(0.5^2) * 1 = 0.25$, which does not converge to $0$. We conclude this estimator is also not consistent.

We conjecture that the consistency of importance weighted approaches depends on continuity of the covariates as a function of space and selection of the bandwidth parameter. We expect that the bandwidth parameter would have to be selected in an adaptive way for consistency to hold.

# D  Proof of Consistency of Point Estimation for our Method

In this section, we prove Theorem 1, which shows that our point estimate is consistent under infill asymptotics. We first state a complete version of Theorem 1 that includes an explicit definition for the adaptive choice of neighbors.

**Theorem D.1.** *Fix any $M \in \mathbb{N}$ and $(S_m^\star)_{m=1}^M$. Let $(S_n)_{n=1}^\infty$ be a sequence of points in $\mathcal{S}$ such that infill asymptotics holds with respect to $(S_m^\star)_{m=1}^M$. Suppose Assumptions 1 to 4 and 7. Choose any positive sequence $(a_t)_{t=1}^\infty$ that tends to $0$. Define the sequence $k_N$ recursively by, $k_1 = 1$ and*

$$k_{N+1} = \begin{cases} k_N + 1 & \max\limits_{\substack{1 \le m \le M \\ 1 \le n \le N+1}} 1\{S_n \text{ is a } k_N + 1 \text{ nearest-neighbor of } S_m^\star \in S_{1:N+1}\}d(S_m^\star, S_n) \le a_{k_N} \\ k_N & otherwise. \end{cases}$$

$$\text{(D.1)}$$

*Then $\widehat{\beta}^{N,k_N} \to \beta^{MLE}$, where convergence is in distribution.*

We first show that the sequence of number of neighbors $(k_N)_{N=1}^\infty$ has two desirable properties. First, it tends to infinity. Second, the maximum distance of the $k_N$ nearest-neighbors to each target in location tends to $0$ as $N$ increases. The first property is needed for the variance of our estimate to tend to $0$, and the second property is ensures that the bias in our point estimate goes to $0$ as $N$ increases.

**Proposition D.1.** *Fix any $M \in \mathbb{N}$ and $(S_m^\star)_{m=1}^M$. Let $(S_n)_{n=1}^\infty$ be a sequence of points in $\mathcal{S}$. Then if $(S_n)_{n=1}^\infty$ satisfies infill asymptotics with respect to $(S_m^\star)_{m=1}^M$. Choose $(a_t)_{t=1}^\infty$ to be any positive sequence tending to $0$. Define the sequence $(k_N)_{N=1}^\infty$ by $k_1 = 1$ and*

$$k_{N+1} = \begin{cases} k_N + 1 & R_{N+1,k_N+1} \le a_{k_N} \\ k_N & otherwise. \end{cases} \quad \text{(D.2)}$$

*with $R_{N,t} = \max\limits_{1 \le m \le M} \max\limits_{1 \le n \le N} 1\{S_n \text{ is a } t \text{ nearest-neighbor of } S_m^\star\}d(S_m^\star, S_n)$ Then the following two properties hold:*

    *1. $\lim_{N\to\infty} k_N = \infty$; and*

    *2. $\lim_{N\to\infty} R_{N,k_N} = 0$.*

*Proof.* We first show that the sequence $(k_N)_{N=1}^\infty$ is unbounded. Because it is monotone increasing, this implies property 1.

Towards contradiction, suppose there exists a least upper bound $K$ such that $k_N \leq K$ for all $N$. Because the $k_N$, we can find a $K$ such that $k_N = K$ for some $N$, and $K$. Because $k_N$ is monotone increasing, it must be the case that for all $N \geq N_0$, $k_N = K$. Therefore, we must have that for all $N \geq N_0$,

$$R_{N+1,K+1} > a_K > 0. \tag{D.3}$$

Otherwise, there would exist an $N'$ such that $k_{N'+1} = K+1$ (by condition 1 in the definition of $k_{N+1}$, contradicting that $K$ is an upper bound on $(k_N)_{n=1}^\infty$. we would have $k_{N'+1} = k_{N'} + 1 = K + 1$.

We now show that there exists a $\tilde{N}$ such that for all $N \geq \tilde{N}, R_{N,K} \leq a_K$ leading to a contradiction. Because infill asymptotics holds, for $1 \leq m \leq M$, there exists a $N_{a_K,m,K}$ such that for all $N \geq N_{a_K,m,K}$, there exists at least $K$ training locations in $B(S_m^\star, a_K)$. Define $\tilde{N} = \max_{1 \leq m \leq M} N_{a_K,m,K}$. Then for all $N \geq \tilde{N}$

$$\max_{1 \leq m \leq M} \max_{1 \leq m \leq N} 1\{S_n \text{ is a } K \text{ nearest-neighbor of } S_m^\star\} d(S_m^\star, S_n) \leq a_K, \tag{D.4}$$

because the $K$ nearest-neighbors of $S_m^\star$ are all contained in $B(S_m^\star, a_K)$ for each $1 \leq m \leq M$. This is a contradiction, leading to the conclusion that no upper bound on $(k_N)_{N=1}^\infty$ exists, and therefore property 1 holds.

It remains to show that property 2 holds. The sequence $(R_{N,k_N})_{N=1}^\infty$ only (possibly) increases between pairs $N, N + 1$ such that $k_{N+1} = k_N + 1$.

For such $N$, $R_{N+1,k_{N+1}} \leq a_{k_N}$. For any $N$ such that $k_N \geq 2$,

$$R_{N+1,k_{N+1}} \leq \max(R_{N,k_N}, a_{k_N}). \tag{D.5}$$

Applying the previous equation to its own right hand side, for any $N$ such that $k_{N-1} \geq 2$,

$$R_{N+1,k_{N+1}} \leq \max(a_{k_N-1}, a_{k_N}). \tag{D.6}$$

Because $(a_t)_{t=1}^\infty$ tends to 0 and $k_N \to \infty$, $\lim_{N \to \infty} \max(a_{k_N-1}, a_{k_N}) = 0$. Therefore, $R_{N,k_N}$ is a non-negative sequence bounded above by a sequence tending to 0, and so $\lim_{N \to \infty} R_{N,k_N} = 0$. $\square$

We now show that the second condition implies the weaker condition that the average distance of the $k_N$ nearest-neighbors to each target location tends to 0 as $N$ increases. This is a useful condition because it implies that the bias in our point estimate goes to 0 as $N$ increases.

**Proposition D.2.** *Let $(k_N)_{N=1}^\infty$ be a sequence of numbers of neighbors such that*

    *1. $\lim_{N \to \infty} k_N = \infty$*

    *2. $\lim_{N \to \infty} \max_{1 \leq m \leq M} \max_{1 \leq n \leq N} 1\{S_n \text{ is a } k_N \text{ nearest-neighbor of } S_m^\star\} d(S_m^\star, S_n) = 0.$*

*Then $\lim_{N \to \infty} \max_{1 \leq m \leq M} \frac{1}{k_N} \sum_{n=1}^N 1\{S_n \text{ is a } k_N \text{ nearest-neighbor of } S_m^\star\} d(S_m^\star, S_n) = 0.$*

*Proof.* By Hölder's inequality

$$\max_{1 \leq m \leq M} \frac{1}{k_N} \sum_{n=1}^N 1\{S_n \text{ is a } k_N \text{ nearest-neighbor of } S_m^\star\} d(S_m^\star, S_n) \tag{D.7}$$

$$\leq \max_{1 \leq m \leq M} \max_{1 \leq n \leq N} 1\{S_n \text{ is a } k_N \text{ nearest-neighbor of } S_m^\star\} d(S_m^\star, S_n). \tag{D.8}$$

The result follows from taking a limit on both sides as $N \to \infty$, using the the left side is nonnegative, and that the right side tends to 0. $\square$

In showing consistency of our point estimate, we rely on the following lemma, which shows that the point estimate $\widehat{\beta}^{N,k_N}$ is a continuous function of the estimator of the conditional expectation $\Psi^{N,k_N} Y$, on an open neighborhood containing of the conditional expectation $\mathbb{E}[Y^\star | S^\star]$.

**Lemma D.1.** *Suppose Assumptions 1 to 4 and 7. Define the map $\tau : \mathbb{R}^M \to \mathbb{R}^P$ by*

$$\tau(A) = \arg \max_{\beta \in \mathbb{R}^P} \sum_{m=1}^M X_m^{\star \mathrm{T}} \beta A_m - \kappa(X_m^{\star \mathrm{T}} \beta). \tag{D.9}$$

*Then $\tau$ is well-defined and continuously differentiable on an open neighborhood containing $\mathbb{E}[Y^\star | S^\star]$.*

*Proof.* Define the function $F : \mathbb{R}^{2P \times P} \to \mathbb{R}^P$, $F(C, \beta) = C - X^{\star \mathrm{T}} \kappa'(X^\star \beta)$. The matrix of partial derivatives of $F$ with respect to $\beta$ evaluated at $\beta^\star$ is $H_\star = X^{\star \mathrm{T}} \Gamma (X^{\star \mathrm{T}} \beta^\star)^{-1} X^\star$ where $\Gamma$ maps an element of $\mathbb{R}^M$ to the diagonal matrix with diagonal entries: $\Gamma(a)_{mm} = \kappa''(a_m)$.

The implicit function theorem Krantz and Parks (2013, Theorem 3.3.1), together with Assumption 7 implies that there exists a (unique) function $\eta$ in an open neighborhood containing $C^\star :=$ $X^{\star \mathrm{T}} \mathbb{E}[Y^\star | S^\star]$ such that for all $C$ in this open neighborhood $F(C, \eta(C)) = 0$.. Furthermore, because the log-likelihood is smooth, $\eta$ is continuously differentiable in an open neighborhood containing $C^\star$. By construction $\eta(C^\star) = \beta^\star$.

Define $\tau(A) = \eta(X^{\star \mathrm{T}} A)$ for all $A \in \mathbb{R}^M$. Let $U_{C^\star}$ be an open neighborhood containing $C^\star$, such that $\eta$ is well-defined, continuously differentiable on $U_{C^\star}$ and $F(C, \eta(C)) = 0$ for all $C \in U_{C^\star}$.

The map $\alpha \to X^{\star \mathrm{T}} \alpha$ is continuously differentiable and surjective. Because composition of continuously differentiable functions is continuously differentiable and there exists an open neighborhood $V \subset \mathbb{R}^M$ such that $X^{\star \mathrm{T}} V \subset U_{C^\star}$ and so $\tau$ is well-defined and continuously differentiable on an open set containing $\mathbb{E}[Y^\star | S^\star]$.

It remains to show that there is an open neighborhood containing $\mathbb{E}[Y^\star | S^\star]$ such that $\tau(A) = \arg\max_{\beta \in \mathbb{R}^P} \sum_{m=1}^M X_m^{\star \mathrm{T}} \beta A_m - \kappa(X_m^{\star \mathrm{T}} \beta)$. The definition of $\eta$ implies that, $F(C, \eta(C)) = 0$ for all $C$ in an open neighborhood of $C^\star$. This in turn implies that for all $A$ in an open neighborhood of $\mathbb{E}[Y^\star | S^\star]$,

$$F(X^{\star \mathrm{T}} A, \eta(X^{\star \mathrm{T}} A)) = F(X^{\star \mathrm{T}} A, \tau(X^{\star \mathrm{T}} A)) = X^{\star \mathrm{T}} A - X^{\star \mathrm{T}} \kappa'(X^\star \tau(X^{\star \mathrm{T}} A)) = 0. \quad \text{(D.10)}$$

This is the first order optimality condition for the maximum in Eq. (D.9). To check second order optimality, we can inspect the Hessian — which only depends on $A$ through the value of $\tau(A)$. This is strictly positive definite in $\beta$ for all $A$ in an open neighborhood of $\mathbb{E}[Y^\star | S^\star]$, as it is strictly positive definite in a neighborhood of $\beta^\star$ by Assumption 7, and because we have already shown $\tau$ is continuous. $\qquad \square$

The second main ingredient in the proof of Theorem 1 is the following lemma, which shows that the empirical conditional expectation converges to the true conditional expectation in distribution.

**Lemma D.2.** *Suppose Assumptions 1 to 4 and 7. Let $(k_N)_{N=1}^\infty$ be any sequence of numbers of neighbors such that*

    *1. $\lim_{N \to \infty} k_N = \infty$*

    *2. $\lim_{m \to \infty} \max_{1 \le m \le M} \frac{1}{k_N} \sum_{n=1}^N 1\{S_n \text{ is a } k_N \text{ nearest-neighbor of } S_m^\star\} d(S_m^\star, S_n) \to 0$.*

*Then $\Psi^{N, k_N} Y_N \to \mathbb{E}[Y^\star | S^\star]$ in distribution, where $\Psi^{N, k_N}$ is the $k_N$ nearest-neighbor weight matrix defined in Definition 2.*

*Proof.* The proof has two steps. First, we show that the expected value of the estimator converges to $\mathbb{E}[Y^\star | S^\star]$. This uses the second property of the sequence of number of neighbors $(k_N)_{N=1}^\infty$ together with Assumption 3. Second, we use a weak law of large numbers to show that the empirical conditional expectation converges in distribution to its expected value.

*Step 1.* We first show that $\mathbb{E}[\Psi^{N, k_N} Y_N | S_1, \dots, S_N] \to \mathbb{E}[Y^\star | S^\star]$. By the definition of $\Psi^{N, k_N}$

$$\mathbb{E}[\Psi^{N, k_N} Y_N | S_1, \dots, S_N] = \frac{1}{k_N} \sum_{m=1}^M \sum_{n=1}^N 1\{S_n \text{ is a } k_N \text{ nearest-neighbor of } S_m^\star\} \mathbb{E}[Y_N | S_1, \dots, S_N].$$
$$\text{(D.11)}$$

By Assumption 2 and Assumption 3 for any $1 \le m \le M$,

$$|\mathbb{E}[(\Psi^{N, k_N} Y_N)_m | S_1, \dots S_N] - \mathbb{E}[Y_m^\star | S_m^\star]| \qquad\qquad\qquad\qquad\qquad \text{(D.12)}$$

$$= \left| \frac{1}{k_N} \sum_{n=1}^N 1\{S_n \text{ is a } k_N \text{ nearest-neighbor of } S_m^\star\}(f(S_n) - f(S_m^\star)) \right| \quad \text{(D.13)}$$

$$\le \frac{L}{k_N} \sum_{n=1}^N 1\{S_n \text{ is a } k_N \text{ nearest-neighbor of } S_m^\star\} d(S_m^\star, S_n). \qquad \text{(D.14)}$$

By the second property of $(k_N)_{N=1}^\infty$

$$\lim_{N \to \infty} \max_{1 \le m \le M} \frac{1}{k_N} \sum_{n=1}^N 1\{S_n \text{ is a } k_N \text{ nearest-neighbor of } S_m^\star\} d(S_m^\star, S_n) = 0. \tag{D.15}$$

Therefore,

$$\lim_{N \to \infty} \max_{1 \le m \le M} \left| \mathbb{E}[(\Psi^{N,k_N} Y_N)_m | S_1, \dots S_N] - \mathbb{E}[Y_m^\star | S_m^\star] \right| = 0. \tag{D.16}$$

We next show that $\Psi^{N,k_N} Y_N \to \mathbb{E}[Y^\star | S^\star]$ in distribution. For this we use a weak law of large numbers for triangular arrays. Centering gives us,

$$(\Psi^{N,k_N} Y_N)_m = \frac{1}{k_N} \sum_{n=1}^N 1\{S_n \text{ is a } k_N \text{ nearest-neighbor of } S_m^\star\}(Y_n - \mathbb{E}[Y_n | S_n]) \tag{D.17}$$

$$+ \mathbb{E}[(\Psi^{N,k_N} Y_N)_m | S_1, \dots S_N]. \tag{D.18}$$

The random variables $Y_n - \mathbb{E}[Y_n | S_n]$ have mean 0. For each $1 \le m \le M$, $N \in \mathbb{N}$ and $1 \le n \le N$, define

$$\tilde{Y}_{n,m}^N = \frac{1}{k_N} 1\{S_n \text{ is a } k_N \text{ nearest-neighbor of } S_m^\star\}(Y_n - \mathbb{E}[Y_n | S_n]). \tag{D.19}$$

For $N \in \mathbb{N}$. The conditional variance of the partial sums is

$$\mathbb{V}[\sum_{n=1}^N \tilde{Y}_{n,m}^N] = \frac{1}{k_N^2} \sum_{n=1}^N 1\{S_n \text{ is a } k_N \text{ nearest-neighbor of } S_m^\star\} \mathbb{E}[(Y_n - \mathbb{E}[Y_n | S_n])^2 | S_n] \tag{D.20}$$

$$\le \frac{B_Y}{k_N}. \tag{D.21}$$

The inequality follows from Assumption 4 and the fact that

$$\sum_{n=1}^N 1\{S_n \text{ is a } k_N \text{ nearest-neighbor of } S_m^\star\} = k_N. \tag{D.22}$$

Therefore, for each $1 \le m \le M$, the sequence $(\tilde{Y}_{n,m}^N)_{n=1}^N$ is a triangular array of independent random variables with mean 0 and variance bounded by $\frac{B_Y}{k_N}$. By the first property of the $(k_N)_{N=1}^\infty$ sequence, $\mathbb{V}(\sum_{n=1}^N \tilde{Y}_{n,m}^N) \to 0$ as $N \to \infty$. By Chebyshev's inequality

$$\mathbb{P}\left( \left| \frac{1}{k_N} \sum_{n=1}^N 1\{S_n \text{ is a } k_N \text{ nearest-neighbor of } S_m^\star\}(Y_n - \mathbb{E}[Y_n | S_n]) \right| > \epsilon \right) \le \frac{B_Y}{k_N \epsilon^2}. \tag{D.23}$$

Because $\frac{B_Y}{k_N \epsilon^2} \to 0$ as $N \to \infty$

$$\frac{1}{k_N} \sum_{n=1}^N 1\{S_n \text{ is a } k_N \text{ nearest-neighbor of } S_m^\star\}(Y_n - \mathbb{E}[Y_n | S_n]) \to 0 \tag{D.24}$$

in distribution for each $1 \le m \le M$. Therefore, $(\Psi^{N,k_N} Y_N)_m \to \mathbb{E}[Y_m^\star | S_m^\star]$ in distribution for each $1 \le m \le M$. $\qquad\square$

We now show that for any sequence $(k_N)_{N=1}^\infty$ that satisfies the two properties described in Proposition D.2, our point estimate $\widehat{\beta}^{N,k_N}$ converges in distribution to the maximum likelihood parameter $\beta^{\text{MLE}}$.

**Theorem D.2.** *Suppose Assumptions 1 to 4 and 7. Let $(k_N)_{n=1}^N$ be chosen as in Theorem 1. Then $\widehat{\beta}^{N,k_N} \to \beta^{\text{MLE}}$, where convergence is in distribution.*

*Proof of Theorem 1.* Proposition D.1 and Proposition D.2 imply the selected $k_n$ satisfy the assumptions of Lemma D.2, and so

$$\Psi^{N,k_N} Y_N \to \mathbb{E}[Y^\star | S^\star] \tag{D.25}$$

in distribution. By Lemma D.1, the map $\tau$ is continuous on an open neighborhood containing $\mathbb{E}[Y^\star | S^\star]$. The continuous mapping theorem implies

$$\widehat{\beta}^{N,k_N} = \tau(\Psi^{N,k_N} Y_N) \to \tau(\mathbb{E}[Y^\star | S^\star]) = \beta^{\text{MLE}} \tag{D.26}$$

in distribution. $\qquad\square$

 # E Proof of Asymptotic Validity of Confidence Intervals

601 In this section, we prove Theorem 2. We first prove a lemma that states that, for large $N$, the
602 nearest-neighbor sets used in estimation are disjoint for each $m$. This simplifies our analysis, as
603 many of the sums involved then consist of independent random variables. We then show that our
604 variance estimate is consistent, and that our stated bound on the bias is an upper bound on a consistent
605 estimate of the bias. Next, we prove asymptotic normality of our estimate of $\mathbb{E}[Y^\star|S^\star]$. Finally, we
606 use the delta method to prove asyptotic normality of our estimator, and combine this with our earlier
607 consistency results for the moments to show Theorem 2.

## E.1 Preliminary Results

609 We first show the following lemma, which will be used in several subsequent results. It states that for
610 large $N$, the nearest-neighbor sets used for estimating $\mathbb{E}[Y^\star|S^\star]$ are disjoint.

611 **Lemma E.3.** *Let $(S_n)_{n=1}^N$ be a sequence of points in $\mathcal{S}$ such that infill asymptotics holds with respect*
612 *to $(S_m^\star)_{m=1}^M$. Suppose that $k_N$ is chosen according to Theorem 1. Then there exists an $N_0$ such that*
613 *for all $N \geq N_0$, and all $1 \leq m, m' \leq M$ with $m \neq m'$ and $1 \leq n \leq N$, $\Psi_{mn}^{N,k_N} \Psi_{m'n}^{N,k_N} = 0$.*

614 *Proof.* Because all the $(S_m^\star)_{m=1}^M$ are distinct we can find an $\epsilon > 0$ such that for all $1 \leq m, m' \leq M$,
615 $m \neq m'$, we have that $d_\mathcal{S}(S_m^\star, S_{m'}^\star) > 2\epsilon$. Proposition D.1, property 2 implies that there exists
616 an $N_0$ such that for all $N \geq N_0$ and all $1 \leq m \leq M$, if $S_n$ is a $k_N$ nearest-neighbor of $S_m^\star$, then
617 $d_\mathcal{S}(S_n, S_m^\star) < \epsilon$. For all $1 \leq m, m' \leq M$ with $m \neq m'$ and any $1 \leq n \leq N$ the triangle inequality
618 states

$$d_\mathcal{S}(S_n, S_m^\star) + d_\mathcal{S}(S_n, S_{m'}^\star) \geq d_\mathcal{S}(S_m^\star, S_{m'}^\star) > 2\epsilon. \tag{E.1}$$

619 Therefore either $d_\mathcal{S}(S_n, S_m^\star) > \epsilon$ or $d_\mathcal{S}(S_n, S_{m'}^\star) > \epsilon$. This implies that for all $N \geq N_0$, $S_n$
620 cannot be a $k_N$ nearest-neighbor of both $S_m^\star$ and $S_{m'}^\star$. We conclude that for all $N \geq N_0$, and all
621 $1 \leq m, m' \leq M$ with $m \neq m'$ and $1 \leq n \leq N$, $\Psi_{mn}^{N,k_N} \Psi_{m'n}^{N,k_N} = 0$. $\square$

622 We next show that one point cannot be the nearest-neighbor of many other points in Euclidean
623 space. This is a key lemma that will be used in the our proof of consistency of our variance estimate.
624 Lemma E.5. It us used to show that the estimate of the variance does not place too much weight on
625 any single observation.

626 **Lemma E.4.** *Let $A \subset \mathbb{R}^d$ a finite set. For any $p \in A$, define the set*

$$A_p := \{a \in A : d(a, p) = \min_{a' \in A} d(a, a')\}. \tag{E.2}$$

627 *Then $|A_p| \leq H_d$ where $H_d$ is a constant that is independent of the set $A$ and the point $p$.*

628 *Proof.* For a point $p$ and a set $A$, let $A - \{p\} = \{a - p : a \in A\}$. Then, $A_p = (A - \{p\})_0$. As the
629 set $A$ is an arbitrary finite set in our statement, we may assume $p = 0$ without loss of generality.

630 We can restrict to cases where $|A_0| \geq 2$. Otherwise the constant $H_d = 2$ suffices. In the case,
631 $|A_0| \geq 2$, let $a, a' \in A_0$ be distinct points. Without loss of generality, we assume that $\|a\| \leq \|a'\|$
632 (otherwise rename the points).

633 For any such points, the definition of $A_0$ implies

$$\|a\| \leq \|a - a'\| \quad \text{and} \quad \|a'\| \leq \|a - a'\|. \tag{E.3}$$

634 We will show that this implies that the angle between $a$ and $a'$ cannot be too small. Using the Hilbert
635 space structure of $\mathbb{R}^d$, we can rewrite Eq. (E.3)

$$0 \leq \|a'\|^2 - 2\langle a, a' \rangle \quad \text{and} \quad \|a\|^2 - 2\langle a, a' \rangle. \tag{E.4}$$

636 Define,

$$\theta = \frac{\langle a, a' \rangle}{\|a\|\|a'\|}. \tag{E.5}$$

Expanding the squared distance

$$\|a - a'\|^2 = \|a\|^2 + \|a'\|^2 - 2\theta\|a\|\|a'\|. \tag{E.6}$$

Then

$$\|a\|^2 - 2\theta\|a\|\|a'\| > 0 \tag{E.7}$$

and so, using that $\|a\| \leq \|a'\|$, $\cos(\theta) \leq \frac{1}{2}$. This implies that the normalized vectors $\frac{a}{\|a\|}$ and $\frac{a'}{\|a'\|}$ are at least $60°$ apart, which in turn implies that they are separated by a distance of at least $1$. The number of distinct points satisfying this criterion separation criterion is upper bounded by the $1/2$-packing number of the unit sphere embedded in $\mathbb{R}^d$, which is finite because the sphere is compact. Therefore, there can be at most $H_d$ points in $A_0$, where $H_d$ is the $1/2$-packing number of the unit sphere embedded in $\mathbb{R}^d$. $\qquad\square$

## E.2 Consistency of Variance Estimate

Define the sequence of maps $\zeta^N : \{1, \ldots, N\} \to \{1, \ldots, N\}$ to map $S_n$ to the index of its nearest-neighbor (not equal to itself). We assume that all $S_n$ are distinct, although random tie-breaking can be used otherwise, with some added complexity needed to handle additional probabilistic arguments.

**Lemma E.5.** *Let $(S_n)_{n=1}^N$ be a sequence of points in $\mathbb{R}^d$ such that infill asymptotics holds with respect to $(S_m^\star)_{m=1}^M$. Suppose Assumptions 1 to 6. Then $k_N \Psi^{N,k_N} \Lambda (\Psi^{N,k_n})^{\mathrm{T}} \to \Lambda^\star$, where $\Lambda^N$ is a diagonal matrix with $\Lambda_{nn}^N = \frac{1}{2}(Y_n - Y_{\zeta^N(n)})$ and $\Lambda^\star$ is a diagonal matrix with $\Lambda^\star = \mathbb{V}[Y_m^\star | S_m^\star]$ for $1 \leq m \leq M$ and convergence is in distribution.*

*Proof.* We write entries in the matrix

$$k_N(\Psi^{N,k_N} \Lambda^N (\Psi^{N,k_N})^{\mathrm{T}})_{mm'} = k_N \sum_{n=1}^N \Psi_{mn}^{N,k_N} \Psi_{m'n}^{N,k_N} \frac{1}{2}(Y_n - Y_{\zeta^N(n)})^2. \tag{E.8}$$

By Lemma E.3, for all $N$ sufficiently large, for $m \neq m'$, we have $\Psi_{mn}^{N,k_N} \Psi_{m'n}^{N,k_N} = 0$. Therefore, for all $N$ sufficiently large, $k_N(\Psi^{N,k_N} \Lambda^N (\Psi^{N,k_N})^{\mathrm{T}})_{mm'}$ is diagonal, and we need only consider the entries with $m = m'$.

We expand the quadratic form in Eq. (E.8), and use the identity $\Psi_{mn}^{N,k_N} = k_N(\Psi_{mn}^{N,k_N})^2$

$$k_N(\Psi^{N,k_N} \Lambda^N (\Psi^{N,k_N})^{\mathrm{T}})_{mm} = \underbrace{\frac{1}{2}\sum_{n=1}^N \Psi_{mn}^{N,k_N} Y_n^2}_{:=\Gamma_1} + \underbrace{\frac{1}{2}\sum_{n=1}^N \Psi_{mn}^{N,k_N} Y_{\zeta^N(n)}^2}_{:=\Gamma_2} - \underbrace{\sum_{n=1}^N \Psi_{mn}^{N,k_N} Y_n Y_{\zeta^N(n)}}_{:=\Gamma_2}. \tag{E.9}$$

We will show that the terms $\Gamma_1$ and $\Gamma_2$ each converge to $\frac{1}{2}(\mathbb{V}[Y^\star|S^\star] + \mathbb{E}[Y^\star|S^\star]^2)$, and $\Gamma_3$ converges in distribution to $\mathbb{E}[Y^\star|S^\star]^2$. Given these results, Slutsky's lemma (van der Vaart, 1998, Lemma 2.8), implies completes the proof of the lemma, as each term converges to a constant. For $\Gamma_1, \Gamma_2$ and $\Gamma_3$, the general proof of convergence will be the same: we first show the expectation converges to the claimed value, and then show that the variance converges to $0$. Convergence in distribution is a consequence of the variance tending to $0$ and Chebyshev's inequality.

The expected value of $\Gamma_1$ is

$$\mathbb{E}[\Gamma_1] = \frac{1}{2k_n} \sum_{n=1}^N 1\{S_n \text{ is a } k_N \text{ nearest-neighbor of } S_m^\star\}\mathbb{E}[Y_n^2] \tag{E.10}$$

$$= \frac{1}{2k_n} \sum_{n=1}^N 1\{S_n \text{ is a } k_N \text{ nearest-neighbor of } S_m^\star\}(\mathbb{E}[Y_n]^2 + \mathbb{V}[Y_n]). \tag{E.11}$$

Proposition D.1, property 2, implies that $d(S_n, S_m^\star) \to 0$ for all terms such that $1\{S_n \text{ is a } k_N \text{ nearest-neighbor of }\} \neq 0$. Using continuity of the mean and variance of the response (Assumptions 3 and 5)

$$\lim_{N \to \infty} \max_{1 \leq n \leq N} 1\{S_n \text{ is a } k_N \text{ near. neigh. of } S_m^\star\}((\mathbb{E}[Y_n]^2 + \mathbb{V}[Y_n]) - (\mathbb{E}[Y_m^\star]^2 + \mathbb{V}[Y_m^\star])) = 0. \tag{E.12}$$

And so

$$\lim_{N \to \infty} \frac{1}{k_n} \sum_{n=1}^{N} 1\{S_n \text{ is a } k_N \text{ nearest-neighbor of } S_m^\star\}(\mathbb{E}[Y_n]^2 + \mathbb{V}[Y_n]) = \mathbb{E}[Y_m^\star]^2 + \mathbb{V}[Y_m^\star]. \tag{E.13}$$

We next verify that the variance of $\Gamma_1$ tends to $0$. Because the $Y_n$ are independent

$$\mathbb{V}\Big[\frac{1}{2} \sum_{n=1}^{N} \Psi_{nm}^{N,k_N} Y_n\Big] = \frac{1}{4k_n^2} \sum_{n=1}^{N} 1\{S_n \text{ is a } k_N \text{ nearest-neighbor of } S_m^\star\}\mathbb{V}[Y_n^2]. \tag{E.14}$$

Assumption 3 implies that within an open neighborhood of any of the test locations, $\mathbb{E}[Y_n]$ is uniformly bounded. Combining this with Assumptions 4 and 6 for $N$ sufficiently large, there exists a constant $K$ such that $1\{S_n \text{ is a } k_N \text{ nearest-neighbor of } S_m^\star\}\mathbb{V}[Y_n^2] \leq 1\{S_n \text{ is a } k_N \text{ nearest-neighbor of } S_m^\star\}K$. Therefore,

$$\lim_{N \to \infty} \mathbb{V}\Big[\Gamma_1\Big] \leq \lim_{N \to \infty} \frac{K}{4k_N} = 0 \tag{E.15}$$

where the last equality used that $\lim_{N \to \infty} k_N = \infty$ (Proposition D.1, property 1).

We now consider $\Gamma_2$ (Eq. (E.9)). Because $S_{\zeta^N(n)}$ is the nearest-neighbor of $S_n$, $d(S_{\zeta^N(n)}, S_n) \leq d(S_m^\star, S_n) + \min_{n' \neq n} d(S_{n'}, S_m^\star)$ and so

$$d(S_m^\star, S_{\zeta^N(n)}) \leq d(S_m^\star, S_n) + d(S_{\zeta^N(n)}, S_n) = 2d(S_m^\star, S_n) + \min_{n' \neq n} d(S_{n'}, S_m^\star). \tag{E.16}$$

By the infill assumption and Proposition D.1, property 2,

$$\lim_{N \to \infty} 1\{S_n \text{ is a } k_N \text{ nearest-neighbor of } S_m^\star\}(2d(S_m^\star, S_n) + \min_{n' \neq n} d(S_{n'}, S_m^\star)) = 0. \tag{E.17}$$

We can now apply the same argument as we used for $\Gamma_1$ to show the expectation of $\Gamma_2$ converges:

$$\mathbb{E}[\Gamma_2] = = \frac{1}{2} \sum_{n=1}^{N} \Psi_{nm}^{N,k_N} (\mathbb{E}[Y_{\zeta(n)}]^2 + \mathbb{V}[Y_{\zeta^N(n)}]). \tag{E.18}$$

Now using Assumption 3, Assumption 5 and that $d(S_{\zeta^N(n)}, S_m^\star) \to 0$, for all terms such that $\Psi_{nm}^{N,k_N} \neq 0$,

$$\lim_{N \to \infty} \frac{1}{2} \sum_{n=1}^{N} \Psi_{nm}^{N,k_N} (\mathbb{E}[Y_{\zeta(n)}]^2 + \mathbb{V}[Y_{\zeta^N(n)}]) = \frac{1}{2}(\mathbb{E}[Y^\star]^2 + \mathbb{V}[Y^\star]).$$

We now show the variance of $\Gamma_2$ tends to $0$.

$$\frac{1}{2} \sum_{n=1}^{N} \Psi_{nm}^{N,k_N} Y_{\zeta(n)}^2 = \frac{1}{2} \sum_{n'=1}^{N} \Big(\sum_{n=1}^{N} \Psi_{nm}^{N,k_N} 1\{n' = \zeta^N(n)\}\Big) Y_{n'}^2. \tag{E.19}$$

This is a sum of independent terms. We define the weights

$$a_{n',m}^N = \Big(\frac{1}{2} \sum_{n=1}^{N} \Psi_{nm}^{N,k_N} 1\{n' = \zeta^N(n)\}\Big). \tag{E.20}$$

Then,

$$\mathbb{V}\Big[\frac{1}{2} \sum_{n=1}^{N} \Psi_{nm}^{N,k_N} Y_{\zeta^N(n)}^2\Big] = \sum_{n'=1}^{N} (a_{n',m}^N)^2 \mathbb{V}[Y_{n'}^2] \tag{E.21}$$

From the definition of $a_{n',m}^N$, and using Lemma E.4

$$\sum_{n'=1}^{N} (a_{n',m}^N)^2 = \frac{1}{4}\left( \sum_{n=1}^{N} \Psi_{nm}^{N,k_N} \sum_{r=1}^{N} \Psi_{rm}^{N,k_N} 1\{r = \zeta^N(n)\} \right) \tag{E.22}$$

$$\leq \frac{1}{4k_N}\left( \sum_{n=1}^{N} \Psi_{nm}^{N,k_N} H_d \right) \tag{E.23}$$

$$\leq \frac{H_d}{4k_N}. \tag{E.24}$$

Also, for any open neighborhood containing $S_m^\star$, for all $N$ sufficiently large $a_{n'}^N = 0$ unless $S_{n'}$ is contained in this open neighborhood, so that for terms with non-zero coefficient $\mathbb{V}[Y_{n'}^2]$ is uniformly bounded by some constant $K$ by combining Assumptions 3, 4 and 6. Therefore, for all $N$ sufficiently large, $\sum_{n'=1}^{N} (a_{n'}^N)^2 \mathbb{V}[Y_{n'}^2] \leq \frac{H_d K}{4k_N}$ which tends to 0 because $k_N \to \infty$ (Proposition D.1, property 1).

We consider $\Gamma_3$ (Eq. (E.9)).

$$\sum_{n=1}^{N} \Psi_{mn}^{N,k_N} \mathbb{E}[Y_n Y_{\zeta^N(n)}] = \sum_{n=1}^{N} \Psi_{mn}^{N,k_N} \mathbb{E}[Y_n]\mathbb{E}[Y_{\zeta^N(n)}]. \tag{E.25}$$

Because $\mathbb{E}[Y_n], \mathbb{E}[Y_{\zeta^N(n)}] \to \mathbb{E}[Y_m^\star]$ for all $n$ such that $\Psi_{mn}^{N,k_N} \neq 0$, this converges to $\mathbb{E}[Y_m^\star]^2$. It remains to show that the variance of $\Gamma_3$ converges 0. We expand into variances and covariances,

$$\mathbb{V}[\sum_{n=1}^{N} \Psi_{mn}^{N,k_N} Y_n Y_{\zeta(n)}] = \sum_{n'=1}^{N} \sum_{n=1}^{N} \Psi_{mn}^{N,k_N} \Psi_{mn'}^{N,k_N} \mathrm{Cov}(Y_n Y_{\zeta(n)}, Y_{n'} Y_{\zeta(n')}). \tag{E.26}$$

We can upper bound the covariance term as,

$$\left| \mathrm{Cov}(Y_n Y_{\zeta(n)}, Y_{n'} Y_{\zeta(n')}) \right| \tag{E.27}$$
$$\leq (1\{n = n'\} + 1\{n = \zeta(n')\} + 1\{n' = \zeta(n)\} + 1\{\zeta(n) = \zeta(n')\}) \max_{1 \leq n \leq N} \mathbb{V}(Y_n Y_{\zeta(n)}). \tag{E.28}$$

Because $Y_n, Y_{\zeta(n)}$ are independent,

$$\mathbb{V}(Y_n Y_{\zeta(n)}) = \mathbb{V}(Y_n)\mathbb{V}(Y_{\zeta(n)}) + \mathbb{V}(Y_n)\mathbb{E}[Y_{\zeta(n)}]^2 + \mathbb{V}(Y_{\zeta(n)})\mathbb{E}[Y_n]^2. \tag{E.29}$$

This is bounded by a constant in a region containing the training locations by Assumptions 3 and 4. Call this constant $\gamma$. Then,

$$\mathbb{V}[\sum_{n=1}^{N} \Psi_{mn}^{N,k_N} Y_n Y_{\zeta(n)}] \tag{E.30}$$

$$\leq \gamma \sum_{n=1}^{N} \sum_{n'=1}^{N} \Psi_{mn}^{N,k_N} \Psi_{mn'}^{N,k_N} \left(1\{n = n'\} + 1\{n = \zeta(n')\} + 1\{n' = \zeta(n)\} + 1\{\zeta(n) = \zeta(n')\}\right). \tag{E.31}$$

We now count the number of non-zero terms in this double sum and show that it is $O(k_N)$. The indicator $n = n'$ contributes exactly $k_N$ non-zero terms; Lemma E.4 implies the indicators $1\{n = \zeta(n')\}, 1\{n' = \zeta(n)\}$ contribute at most $H_d k_N$. Finally,

$$\sum_{n=1}^{N} \sum_{n'=1}^{N} \Psi_{mn}^{N,k_N} \Psi_{mn'}^{N,k_N} 1\{\zeta^N(n) = \zeta^N(n')\} \tag{E.32}$$

$$= \sum_{r=1}^{N} 1\{\exists n : r = \zeta^N(n)\} \sum_{n=1}^{N} \sum_{n'=1}^{N} \Psi_{mn}^{N,k_N} \Psi_{mn'}^{N,k_N} 1\{\zeta^N(n) = r\}1\{\zeta^N(n') = r\} \tag{E.33}$$

$$= \sum_{r=1}^{N} 1\{\exists n : r = \zeta^N(n)\} \left( \sum_{n=1}^{N} \Psi_{mn}^{N,k_N} 1\{\zeta^N(n) = r\} \right)^2. \tag{E.34}$$

The total number of $r$ that are nearest-neighbors to a point that is a $k_N$ nearest-neighbor of $S_m^\star$ cannot exceed $k_N$. And $\left(\sum_{n=1}^N \Psi_{mn}^{N,k_N} 1\{\zeta^N(n) = r\}\right)^2 \leq \frac{H_d}{k_N}$. Therefore, this final sum is $O(1/k_N)$. We conclude the variance of $\Gamma_3$ converges to zero as $N$ tends to infinity. $\qquad\square$

### E.3 Asymptotic Normality of Estimate of Conditional Expectation

We begin by proving that the estimate of the conditional expectation $\Psi^{N,k_N} Y$ is asymptotically normal. We first recall the Lyapunov central limit theorem for triangular arrays.

**Theorem E.1** (Lyapunov Central Limit Theorem, Theorem 27.3 Billingsley 1995)**.** *Let* $\{Z_{n1}, \ldots, Z_{nt_n}\}$ *be independent random variables for each $n \in \mathbb{N}$, with*

$$\mu_{nt} = \mathbb{E}[Z_{nt}], \qquad \sigma_{nt}^2 = \mathbb{V}[Z_{nt}], \qquad s_n^2 = \sum_{t=1}^{t_n} \sigma_{nt}^2.$$

*Assume $s_n^2 \to \infty$ and $s_n > 0$ for all $n$. Suppose there exists $\delta > 0$ such that the Lyapunov condition holds:*

$$\lim_{N \to \infty} \frac{1}{s_n^{2+\delta}} \sum_{t=1}^{t_n} \mathbb{E}\big[\,|Z_{nt} - \mu_{nk}|^{2+\delta}\big] = 0.$$

*Then*

$$\frac{\sum_{t=1}^{t_n}(Z_{nt} - \mu_{nt})}{s_n} \to \mathcal{N}(0,1).$$

*That is, the normalized sum converges in distribution to a standard normal random variable.*

We now prove the following lemma, which involves verifying the Lyapunov condition for entries of $\sqrt{k_N}\Psi^{N,k_N}(Y - \mathbb{E}[Y|S])$.

**Lemma E.6.** *Let $(S_n)_{n=1}^N$ be a sequence of points in $\mathcal{S}$ such that infill asymptotics holds with respect to $(S_m^\star)_{m=1}^M$. Suppose that $k_N$ is chosen according to Theorem 1. Suppose Assumptions 1 to 4 and 7 Then,*

$$\lim_{N \to \infty} \sqrt{k_N}\Psi^{N,k_N}(Y - \mathbb{E}[Y|S]) = \mathcal{N}(0, \Lambda^\star) \tag{E.35}$$

*where $\Lambda^\star$ is a diagonal matrix with $\Lambda_{mm}^\star = \mathbb{V}[Y_m^\star|S_m^\star]$ for $1 \leq m \leq M$.*

*Proof.* By Lemma E.3, for $N$ sufficiently large, the rows of $\Psi^{N,k_N}$ are disjoint. Therefore, the entries of $\Psi^{N,k_N}(Y - \mathbb{E}[Y|S])$ are independent for sufficiently large $N$, and so it suffices to show that each entry of the vector $\sqrt{k_N}\Psi^{N,k_N}(Y - \mathbb{E}[Y|S])$ converges in distribution to a univariate normal random variable.

Let $R_m^N = (\sqrt{k_N}\Psi^{N,k_N}(Y - \mathbb{E}[Y|S]))_m$ be the $m$th entry of the vector $\sqrt{k_N}\Psi^{N,k_N}(Y - \mathbb{E}[Y|S])$, and define $r_{nm}^N = \sqrt{k_N}\Psi_{nm}^{N,k_N}(Y - \mathbb{E}[Y|S])$, so that $R_m^N = \sum_{n=1}^N r_{nm}^N$. The variance of $R_m^N$ is

$$\mathbb{V}[R_m^N] = k_N \sum_{n=1}^N (\Psi_{mn}^{N,k_N})^2 \mathbb{V}[Y_n|S_n] = \sum_{n=1}^N \Psi_{mn}^{N,k_N} \mathbb{V}[Y_n|S_n]. \tag{E.36}$$

Assumption 5 and Proposition D.1 imply

$$\sum_{n=1}^N \Psi_{mn}^{N,k_N} \mathbb{V}[Y_n|S_n] \to \mathbb{V}[Y_m^\star|S_m^\star]. \tag{E.37}$$

If $\mathbb{V}[Y_m^\star|S_m^\star] = 0$, then $\mathbb{V}[R_m^N] \to 0$ and so $R_m^N \to 0$ in distribution, as claimed in this case. Otherwise, we consider the limit

$$\lim_{N \to \infty} \frac{1}{\mathbb{V}[R_m^N]^4} \sum_{n=1}^N \mathbb{E}[|r_{nm}^N|^4] \tag{E.38}$$

$$= \lim_{N \to \infty} \frac{1}{(\sum_{n=1}^N \Psi_{mn}^{N,k_N} \mathbb{V}[Y_n|S_n])^4} \sum_{n=1}^N \mathbb{E}[|\sqrt{k_N}\Psi_{nm}^{N,k_N}(Y - \mathbb{E}[Y|S])|^4] \tag{E.39}$$

$$= \lim_{N \to \infty} \frac{1}{k_N^2(\sum_{n=1}^N \Psi_{mn}^{N,k_N} \mathbb{V}[Y_n|S_n])^4} \sum_{n=1}^N \Psi_{nm}^{N,k_N} \mathbb{E}[|(Y_n - \mathbb{E}[Y_n|S])|^4] \tag{E.40}$$

727 Assumption 6 implies $\sum_{n=1}^{N} \Psi_{nm}^{N,k_N} \mathbb{E}[|(Y_n - \mathbb{E}[Y_n|S])|^4] \leq C$, and since $\sum_{n=1}^{N} \Psi_{mn}^{N,k_N} \mathbb{V}[Y_n|S_n] \to$
728 $\mathbb{V}[Y_m^\star|S_m^\star] \neq 0$ the Lyapunov condition holds. $\qquad\square$

729 **Proposition E.1.** *Let $(S_n)_{n=1}^{N}$ be a sequence of points in $\mathcal{S}$ such that infill asymptotics holds with*
730 *respect to $(S_m^\star)_{m=1}^{M}$. Suppose that $k_N$ is chosen according to Theorem 1 with $a_t = \frac{1}{\sqrt{t}}$. Suppose*
731 *Assumptions 1 to 4 and 7 Then,*

$$\sqrt{k_N}(\Psi^{N,k_N}Y - \mathbb{E}[Y^\star|S^\star]) \to \mathcal{N}(B, \Lambda^\star), \tag{E.41}$$

732 *for $B \in \mathbb{R}^M$ with $B_m = \sqrt{k_N}\left(\sum_{n=1}^{N} \Psi^{N,k_N} f(S_n) - f(S_m^\star)\right)$ and $\Lambda^\star$ is a diagonal matrix with*
733 *$\Lambda_{mm}^\star = \mathbb{V}[Y_m^\star|S_m^\star]$ for $1 \leq m \leq M$.*

734 *Proof.* Adding zero,

$$\sqrt{k_N}(\Psi^{N,k_N}Y - \mathbb{E}[Y^\star|S^\star]) = \sqrt{k_N}(\Psi^{N,k_N}Y - \mathbb{E}[Y|S]) + \sqrt{k_N}(\Psi^{N,k_N}(\mathbb{E}[Y|S] - \mathbb{E}[Y^\star|S^\star]). \tag{E.42}$$

735 Lemma E.6 implies that $\sqrt{k_N}(\Psi^{N,k_N}Y - \mathbb{E}[Y|S]) \to \mathcal{N}(0, \Lambda^\star$. Considering the second term,
736 For all $k_N > 2$,

$$\left| \sqrt{k_N}\left(\sum_{n=1}^{N} \Psi^{N,k_N} f(S_n) - f(S_m^\star)\right) \right| \leq L\left(\sum_{n=1}^{N} \Psi^{N,k_N} d(S_n, S_m^\star)\right) \tag{E.43}$$

$$\leq L\left(\sum_{n=1}^{N} \Psi^{N,k_N} d(S_n, S_m^\star)\right) \tag{E.44}$$

$$\leq \frac{Lk_N}{\sqrt{k_N}} a_{k_N-1}. \tag{E.45}$$

737 Because $a_t = \frac{1}{\sqrt{t}}$, $\frac{Lk_N}{\sqrt{k_N}} a_{k_N-1} \leq 2L$. This implies that this bias term is $O(1)$. $\qquad\square$

## E.4   Proof of Asymptotic Validity of Confidence Intervals

739 We now prove that the confidence intervals defined in Section 3.1 are asymptotically valid. We first
740 show that the confidence intervals, with linearization around the true parameter, are asymptotically
741 valid. A key lemma along the way is van der Vaart (1998, Theorem 3.1), which is essentially the
742 conclusion of the delta method. We recall this theorem here for convenience.

743 **Lemma E.7** (Delta Method). *Let $\phi$ be a map defined on a subset $D \subset \mathbb{R}^M \to \mathbb{R}^P$ that is differ-*
744 *entiable at $\theta$. Let $T_n$ be random vectors taking values in $D$. If $r_N(T_N - \theta) \to T$ for $(r_N)_{N=1}^{\infty}$ a*
745 *sequence such that $r_n \to \infty$, then $r_N(\phi(T_N) - \phi(\theta)) \to \phi'_\theta(T)$ in distribution.*

746 We apply this lemma together with Lemma E.6 to show that the point estimate $\widehat{\beta}^{N,k_N}$ is asymptotically
747 normal. After that what will remain is to use consistency of the variance estimate to show that using
748 the estimated variance in place of the true variance yields asymptotically valid confidence intervals,
749 and to use consistency of the point estimate to show that linearization around the point estimate
750 instead of the true parameter yields asymptotically valid confidence intervals.

751 **Theorem E.2** (Asymptotic Normality of Point Estimate). *Let $(S_n)_{n=1}^{N}$ be a sequence of points in $\mathcal{S}$*
752 *such that infill asymptotics holds with respect to $(S_m^\star)_{m=1}^{M}$. Suppose Assumptions 1 to 7. Let $\widehat{\beta}^{N,k_N}$*
753 *be the point estimate defined in Eq. (4). Then,*

$$\sqrt{k_N}(\widehat{\beta}^{N,k_N} - \beta^{MLE}) \to \mathcal{N}(\tau'(C^\star)B, \tau'(C^\star)\Lambda^\star \tau(C^\star)^{\mathrm{T}}), \tag{E.46}$$

754 *where $B$ and $\Lambda^\star$ are as in Proposition E.1*

755 *Proof.* We apply Lemma E.7 with $\phi = \tau$ and $T_N = \Psi^{N,k_N}Y$. The point estimate $\widehat{\beta}^{N,k_N}$ is given by
756 $\tau(\Psi^{N,k_N}Y)$. The true parameter $\beta^{MLE}$ is given by $\tau(\mathbb{E}[Y^\star|S^\star])$. Proposition E.1 implies

$$\sqrt{k_N}(\Psi^{N,k_N}Y - \mathbb{E}[Y^\star|S^\star]) \to \mathcal{N}(B, \Lambda^\star). \tag{E.47}$$

Therefore, we can apply the delta method (Lemma E.7) to conclude that

$$\sqrt{k_N}(\widehat{\beta}^{N,k_N} - \beta^{\mathrm{MLE}}) = \sqrt{k_N}(\tau(\Psi^{N,k_N}Y) - \tau(\mathbb{E}[Y^\star|S^\star])) \tag{E.48}$$

$$\to \mathcal{N}(\tau'(C^\star)B, \tau'(C^\star)\Lambda^\star\tau(C^\star)^{\mathrm{T}}), \tag{E.49}$$

as desired. □

From this, we conclude that,

$$\sqrt{k_N}(\widehat{\beta}_p^{N,k_N} - \beta_p^{\mathrm{MLE}}) \to \mathcal{N}(e_p^{\mathrm{T}}\tau'(C^\star)B, e_p^{\mathrm{T}}\tau'(C^\star)\Lambda^\star\tau(C^\star)^{\mathrm{T}}e_p). \tag{E.50}$$

where $e_p$ is the $p$th standard basis vector in $\mathbb{R}^P$. Defining $\sigma_p^2 = e_p^{\mathrm{T}}\tau'(C^\star)\Lambda^\star\tau(C^\star)^{\mathrm{T}}e_p$, we can construct the pivotal quantity

$$Z_p = \frac{\sqrt{k_N}(\widehat{\beta}_p^{N,k_N} - \beta_p^{\mathrm{MLE}} - e_p^{\mathrm{T}}\tau'(C^\star)B)}{\sqrt{\sigma_p^2}} \to \mathcal{N}(0,1). \tag{E.51}$$

This gives us the corollary that, when linearized around the true parameter, the confidence intervals are asymptotically valid.

**Corollary 1** (Asymptotic Validity of Confidence Intervals Linearized Around True Parameter). *Let $(S_n)_{n=1}^N$ be a sequence of points in $\mathcal{S}$ such that infill asymptotics holds with respect to $(S_m^\star)_{m=1}^M$. Suppose Assumptions 1 to 7. Let $\widehat{\beta}^{N,k_N}$ be the point estimate defined in Eq. (4). Then for any $1 \le p \le P$,*

$$\lim_{N\to\infty} \mathbb{P}\left( \widehat{\beta}_p^{N,k_N} - z_{\alpha/2}\sigma_p - \mu \le \beta_p^{MLE} \le \widehat{\beta}_p^{N,k_N} + z_{\alpha/2}\sigma_p - \mu \right) = 1 - \alpha \tag{E.52}$$

*where $\mu_p = e_p^{\mathrm{T}}\tau'(C^\star)B$ and $\sigma_p^2 = e_p^{\mathrm{T}}\tau'(C^\star)\Lambda^\star\tau(C^\star)^{\mathrm{T}}e_p$ for all $1 \le p \le P$.*

Slutsky's lemma implies that we can replace $\mu_p$ and $\sigma_p^2$ with consistent estimates of the true bias and variance.

**Corollary 2** (Asymptotic Validity of Confidence Intervals With Consistent Estimates). *With the same assumptions as in Corollary 1, let $\widehat{\beta}^{N,k_N}$ be the point estimate defined in Eq. (4). Then for any $1 \le p \le P$,*

$$\lim_{N\to\infty} \mathbb{P}\left( \widehat{\beta}_p^{N,k_N} - z_{\alpha/2}\hat{\sigma}_p - \hat{\mu} \le \beta_p^{MLE} \le \widehat{\beta}_p^{N,k_N} + z_{\alpha/2}\hat{\sigma}_p - \hat{\mu} \right) = 1 - \alpha \tag{E.53}$$

*where $\hat{\mu} = e_p^{\mathrm{T}}\tau'(\Psi^{N,k_N}Y)B$ and $\hat{\sigma}^2 = e_p^{\mathrm{T}}\tau'(\Psi^{N,k_N}Y)\Psi^{N,k_N}\Lambda^N(\Psi^{N,k_N})^{\mathrm{T}}\tau'(\Psi^{N,k_N}Y)^{\mathrm{T}}e_p$ for all $1 \le p \le P$.*

The remaining issue is that, we do not know $\hat{\mu}$, because it depends on the unknown function $f$. We can bound it using the same approach as in Burt et al. (2025a).

**Proposition E.2** (Bounding the bias, Burt et al. (2025a, Proposition 12)).

$$|\hat{\mu}| \le L \sup_{f\in\mathcal{F}_1} \left| \sum_{m=1}^M w_m f(S_m^\star) - \sum_{n=1}^N v_n f(S_n) \right|, \tag{E.54}$$

*where $w = \tau'(\Psi^{N,k_N}Y)^{\mathrm{T}}e_p$ and $v = \Psi^{N,k_N}w$ and $\mathcal{F}_1$ is the set of 1-Lipschitz functions. Moreover, this can be computed efficiently by reduction to a 1-Wasserstein distance between empirical measures.*

*Proof.* The bias term $\hat{\mu}$ is given by

$$\hat{\mu} = \sum_{m=1}^M w_m f(S_m^\star) - \sum_{n=1}^N v_n f(S_n). \tag{E.55}$$

If $L = 0$, $f$ is constant and bias is 0. Otherwise, $\frac{1}{L}f \in \mathcal{F}_1$, and the inequality follows from Assumption 3. The second part of the proposition is Burt et al. (2025a, Proposition 12). □

# F Additional Experimental Details for Simulation Studies

## F.1 Baseline Methods

We compare the proposed method with three baselines:

- **Logistic Regression (LR)**: Fit a logistic regression model to the training data and evaluate the confidence intervals on the target data using the standard errors from the model.

- **Logistic Regression with Sandwich Estimator (LR-Sandwich)**: Fit a logistic regression model to the training data and use the sandwich estimator to compute the standard errors for the confidence intervals on the target data.

- **Weighted Logistic Regression (WLR)**: Fit a weighted logistic regression model to the training data, where the weights are determined by the ratio of the kernel density estimates of the covariate distribution in the training and target data. The weights are computed as follows:

$$w_i = \frac{\hat{p}_T(X_i)}{\hat{p}_S(X_i)} \tag{F.1}$$

  where $\hat{p}_T(X_i)$ is the kernel density estimate of the covariate distribution in the target data and $\hat{p}_S(X_i)$ is the kernel density estimate of the covariate distribution in the training data. The kernel density estimates are computed using Gaussian kernels with bandwidths selected using cross-validation. The weighted logistic regression is then fit using the weights $w_i$.

## F.2 Data Generation

**Infill Simulation.** We generate the training locations uniformly on $[-1, 1]^2$. We generate the target locations on $[-\texttt{scale}, \texttt{scale}]^2$ for $\texttt{scale} = \{i/16\}_{i=1}^{16}$. We use a single covariate, $X$ that is equal to the first spatial coordinate. The expected value of the response variable is given by a $1/1 + \exp(-h(X))$, where $h(X)$ is a piecewise linear function,

$$h(X) = \begin{cases} X & \text{if } X < -0.125 \\ 0.875 - X & \text{if } -0.125 \leq X < 0.125 \\ 0.625 + X & \text{if } X \geq 0.125 \end{cases} \tag{F.2}$$

The response is a Bernoulli random variable with success probability given by the expected value. We generate 10000 training data points and 100 target locations. The training and target locations, conditional expectation of the response, and observed are shown in Fig. 1.

Because the logit of the expected response surface is not linear, logistic regression is misspecified. When the target points are primarily between $[-0.125, 0.125]$, the expected response surface is approximately linear, with a negative slope. On the other hand, over the entire domain, the expected response surface increasing, and should have a positive slope. This means that the logistic regression model will be biased, and the bias will depend on the amount of distribution shift between the training and target data. The amount of distribution shift is controlled by the scale parameter, which determines how far the target locations are from zero.

**Extrapolation Simulation.** We generate data as in the previous experiment, except that the target data is now uniformly distributed on $[-j + 1, j + 1] \times [-1, 1]$ for $j \in \{i/16\}_{i=1}^8$. We also define a new function $h(X)$ that is a piecewise linear function with a different slope, defined as follows:

$$h(X) = \begin{cases} X & \text{if } X < 0.875 \\ 0.875 - X & \text{if } X \geq 0.875 \end{cases} \tag{F.3}$$

This function has a positive slope for $X < 0.875$ and a negative slope for $X \geq 0.875$. The expected response surface is given by $1/1 + \exp(-h(X))$, and the response is a Bernoulli random variable with success probability given by the expected value. As before we generate 10000 training points and 100 target points. We repeat the process for 250 datasets.

