# OpenReview forum: "Wrong Model, Right Uncertainty: Spatial Associations for Discrete Data with Misspecification"
_NeurIPS.cc/2025/Workshop/Reliable_ML — NeurIPS 2025 - Reliable ML Workshop_

### Official Review · Reviewer_ap2c · 2025-09-15
**Technically solid and relevant contribution. This paper introduces the first asymptotically conservative CIs for spatial associations with discrete outcomes under misspecification and nonrandom locations with their claims validated by strong theory and clear simulations.**

**Rating:** 7
**Confidence:** 4

**Review:**

**Summary**

The paper targets inference for spatial associations in binary or count-valued data where (i) GLMs are mis-specified and (ii) training/target locations are non‑i.i.d./non‑aligned. It defines the estimand as the target‑location KL‑projection $\beta_{MLE}$ and proposes a consistent estimator under infill asymptotics by adaptively borrowing responses from $k_N$ ​nearest training neighbors at each target and mapping to GLM coefficients via $\tau(\cdot)$. Consistency requires $k_N$ to grow with sample size. For uncertainty, the paper proves asymptotic normality via a Lyapunov CLT and delta method, and constructs asymptotically conservative CIs that add a Lipschitz‑based bias bound, with a nearest‑neighbor variance estimator consistent under heteroskedastic noise. Simulations show that classical baselines can have zero coverage and wrong‑sign intervals, while the proposed intervals consistently achieve coverage at or above nominal level with no wrong‑sign positives. Though the result comes at the cost of wider intervals and lower true positive rate, especially under extrapolation, it focuses on improving the worst-case performance, enhancing reliability on top of existing methods.

**Strengths**

**Well-written & Easy to follow:** The paper is very well written and easy to follow.

**Novel scope:** The paper introduces first CIs for spatial GLM associations under simultaneous misspecification and nonrandom sampling with discrete outcomes.

**Rigorous formulation & Solid theoretic guarantees:** Clear asymptotic guarantees on consistency and coverage are backed by rigorous problem formulation and solid theoretic derivations.

**Compelling failure cases by simulation:** Simulations starkly illustrate how standard intervals can be over‑confident and wrong‑signed, underscoring reliability risks the method addresses, emphasizing the importance and novelty of this work.

**Weaknesses/Limitations**

**Very specific scope:** The paper introduces strict assumptions and focuses on a very specific setting. It is totally understandable as it lays the ground for all the theoretic math work, but still, it will be great if the authors can provide some clarification on why studying on these settings is helpful, and how the assumptions reflect any possible real-life scenario. Also, it will be interesting to see how the proposed method can be generalized to a broader setting (an empirical study itself will be interesting enough).

**Dependency on smoothness constant  $L$**: The CI requires a Lipschitz constant $L$. In simulations $L$ is set to the true value, but its practical estimation in real life is usually nontrivial. Guidance or data‑driven upper bounds would help.

**Infill assumption:** Guarantees rely on training locations densifying near targets. In persistent extrapolation regimes, intervals may be very wide or the estimand poorly identified. Empirical discussion of how to diagnose infill adequacy would be valuable.

**Validation with real data:** As mentioned by the authors in the conclusion, real‑data case studies are missing. Studies on real datasets would showcase applicability and real value of the proposed method, as the simulation focuses on extreme settings artificially put up in preference for the proposed method, which may not exist in real life. The proposed method will be more compelling if it can be used to correct a real-life pitfall made by existing methods.

**Suggestions for authors**

**Add case studies with real datasets:** Include at least one real‑world dataset with binary or count outcomes and spatial covariates to illustrate end‑to‑end usage, and it will also be helpful in clarifying how all the hyperparameters are chosen in practice.

**Extensions to broader settings:** Briefly discuss non‑Euclidean metrics and multiscale smoothness (piecewise‑Lipschitz) common in environmental processes, and how the proposed method can generalize.

---

### Official Review · Reviewer_ZzDh · 2025-09-19

**Rating:** 8
**Confidence:** 3

**Review:**

The paper provides a well-defined setup: estimating an association between a covariate and a binary- or count-valued response. The motivation is well written and the problem setup is well-defined. It provides a detailed comparison with related works, like  Burt et al. (2025a).
The reasoning about the required assumptions, why they are important, is clearly stated.

For the main contribution, the paper constructs confidence intervals that are asymptotically guaranteed to achieve nominal coverage. In the analysis, it applies Lyapunov central limit theorem together with the delta method. Although I'm not quite familiar with the literature, I think the idea is interesting and it leads to meaningful results.

I suggest the authors further polish the introduction of model (section 2.2), the well-specified part is clear. However, what kind of misspecification will be added to the model is not discussed. "Incontrast, when the model is misspecified, maximizing the log-likelihood instead yields the coefficients that minimize the Kullback–Leibler (KL) divergence between the model and the true data-generating". This is the explanation I found, but I feel it insufficient. Besides, the presentation can be improved, by providing more intuitive examples in the background part.

Overall, I think it's a good paper.